# Functional recruitment and connectivity of the cerebellum is associated with the emergence of Theory of Mind in early childhood

Aikaterina Manoli [1,2,3,4,5] ✉, Frank Van Overwalle [6,7],
Charlotte Grosse Wiesmann [3,8,10] & Sofie L. Valk [2,4,9,10] ✉

There is accumulating evidence that the human cerebellum is heavily implicated in adult social cognition. Yet, its involvement in the development of Theory of Mind (ToM), a hallmark of social cognition, remains elusive. Using openly available functional MRI data of children with emerging ToM abilities ($N = 41$, age range: 3-12 years) and adults ($N = 78$), we show that children who pass a false-belief assessment of ToM abilities activate cerebellar Crus I-II in response to ToM events during a movie-watching task, similar to adults. This activation is not statistically significant in children who do not pass the ToM assessment. Functional connectivity profiles between cerebellar and cerebral ToM regions differ as a function of children's ToM abilities. Notably, task-driven connectivity shifts from upstream to downstream connections between cerebellar and cerebral ToM regions from childhood to adulthood. Greater dependence on connections emerging from the cerebellum early in life suggests an important role of the cerebellum in establishing the cognitive processes underlying ToM in childhood and thus for the undisrupted development of social cognition.

Human interaction crucially relies on our ability to infer other people's thoughts and intentions, a process widely known as Theory of Mind (ToM) or mentalizing[1,2]. Recent advancements in functional magnetic resonance imaging (fMRI) have greatly enhanced our neurobiological understanding of ToM, revealing its neural correlates not only in the cerebral cortex[3–5] but also in the adult cerebellum[6–9]. Although substantial evidence highlights the cerebellum's involvement in ToM processing in adults, its role in the development of ToM remains poorly understood. This gap is particularly striking given that cerebellar lesions acquired during development are linked to profound and enduring ToM deficits—far more severe than those observed following lesions sustained in adulthood[10–12]—suggesting a crucial involvement of the developing cerebellum in ToM understanding. To address this issue, the present study investigated the functional involvement of the

[1]International Max Planck Research School on Cognitive Neuroimaging (IMPRS CoNI), Max Planck Institute for Human Cognitive and Brain Sciences, Leipzig, Germany. [2]Lise Meitner Group Neurobiosocial, Max Planck Institute for Human Cognitive and Brain Sciences, Leipzig, Germany. [3]Minerva Fast Track Group Milestones of Early Cognitive Development, Max Planck Institute for Human Cognitive and Brain Sciences, Leipzig, Germany. [4]Institute of Neuroscience and Medicine, Brain & Behavior (INM-7), Research Centre Jülich, Jülich, Germany. [5]Faculty of Medicine, Leipzig University, Leipzig, Germany. [6]Brain, Body and Cognition Research Group, Faculty of Psychology and Educational Sciences, Vrije Universiteit Brussel, Brussels, Elsene, Belgium. [7]Center for Neurosciences (C4N), Vrije Universiteit Brussel, Brussels, Elsene, Belgium. [8]Cognitive Neuroscience Lab, Department of Liberal Arts and Sciences, University of Technology Nuremberg, Nuremberg, Germany. [9]Institute of Systems Neuroscience, Medical Faculty, Heinrich Heine University Düsseldorf, Düsseldorf, Germany. [10]These authors jointly supervised this work: Charlotte Grosse Wiesmann, Sofie L. Valk. ✉e-mail: manoli@cbs.mpg.de; valk@cbs.mpg.de

cerebellum in the development of ToM during early childhood to better understand how this structure is associated with the emergence of ToM abilities.

Despite being traditionally associated with sensorimotor processing[13–15], a surge of findings over the last three decades highlights the importance of the cerebellum for a wide range of cognitive functions, including language and social cognition[6,16–20]. In the context of ToM, the adult posterior cerebellum, especially Crus I–II, is consistently activated during ToM processing, pointing towards a substantial involvement of this region in ToM[6,7,9,21]. This evidence has been enriched by findings of functional and anatomical connectivity of the adult posterior cerebellum with the cerebral ToM network, which spans frontal and temporoparietal regions such as the dorsal and ventral medial prefrontal cortex (d/vmPFC), the temporoparietal junction (TPJ), and the precuneus (PreC)[4,5]. Recent studies have demonstrated bidirectional task-driven connections and white matter loops between the posterior cerebellum and the ToM network[7,22–25], suggesting the importance of the cerebellum for ToM in terms of extensive information exchange with ToM regions of the cerebral cortex.

In development, the emergence of ToM abilities has been extensively studied across populations ranging from infancy to late childhood[26]. A pivotal milestone in the development of ToM reasoning occurs between the ages of 3 and 5, a breakthrough period in which children typically start succeeding in false-belief tasks, widely regarded as a critical test of ToM abilities[27]. These tasks require children to recognize false beliefs held by a story character, typically in the context of the character's mental misrepresentations regarding an object's location, content, or nature[27–29]. Successfully passing false-belief tasks is argued to reflect the emergence of representations of others' mental states[30]. On a neural level, this cognitive breakthrough is supported by a rich spectrum of structural and functional changes within the cerebral ToM network. Success in false-belief tasks during this developmental period is associated with increases in cortical thickness, surface area, and white matter maturation of regions of the ToM network[3,29,31]. These structural changes are accompanied by greater functional specialization and strengthened connectivity among regions of the ToM network as a function of children's emerging ToM abilities[26,29].

Given the extensive involvement of the cerebellum in ToM in adults, it remains unclear whether the cerebellum is associated with the development of ToM abilities in children in a manner comparable to the cerebral ToM network. This question is far from trivial: the cerebellum appears to be particularly important for the typical development of sociocognitive functions. Pediatric evidence indicates that early-life cerebellar injury often results in severe and persistent changes in social behavior, frequently observed in the context of neurodevelopmental and psychiatric disorders such as autism spectrum disorder (ASD) and schizophrenia[11,32–35]. Strikingly, there is a pronounced discrepancy between the effects of cerebellar abnormalities present from early childhood versus lesions acquired in adulthood: early-life cerebellar disruptions lead to dramatic and enduring changes in social behavior, whereas adult-onset damage results in milder social deficits that often resolve more quickly[10]. This highlights the cerebellum's essential role in the development of social cognition and raises the question of its specific involvement in the undisrupted development of ToM abilities.

In the present study, we therefore investigate how the cerebellum is associated with the development of ToM in early childhood, and how this compares to its role in adulthood. To this aim, we leverage openly accessible fMRI data from children (N = 41), whose age range (3–12 years) covers the period of emergence and development of key ToM abilities[26], as well as two adult samples (Richardson et al.[26]: N = 22; Caltech Conte Center[36]: N = 56). All participants watched the same in-scanner movie, and children additionally completed an out-of-scanner ToM assessment requiring them to reason about false beliefs within a

story setting[26]. We identify regions in the posterior cerebellum (Crus I–II) that are consistently activated during movie scenes involving mental states, both in children who succeed in the false-belief assessment and in adults. These regions are not significantly activated in children who do not yet pass the false-belief assessment. We further show that functional connectivity between these cerebellar regions and cerebral ToM areas increases as a function of children's ToM abilities, as measured by the false-belief task. Finally, we examine the directionality of these connections using dynamic causal modeling (DCM), a technique that estimates the influence ToM-related clusters in the cerebellum and cerebral cortex exert on one another[25,37,38]. We demonstrate an inverse pattern of cerebro-cerebellar connectivity between children who passed the false-belief assessment and adults: while children show upstream connections from cerebellar ToM regions to the cerebral ToM network, adults exhibit downstream connections in the opposite direction.

## Results

### Functional recruitment of cerebellar Crus I–II is associated with the emergence of ToM early in life

We first aimed to identify clusters that are generally activated during ToM processing in the developing cerebellum, regardless of children's ToM abilities. To achieve this, we contrasted activation during movie scenes depicting mental states and bodily pain of the characters with a one-sample $t$-test in the entire developmental sample (N = 41). We identified three activation clusters in the posterior cerebellum that showed stronger activation for the in-scanner movie scenes related to mental states than bodily pain of the characters, specifically in right (r) and left (l) lobule IX (rIX: 2 -46 -44; lIX: -8 -46 -44) and the dorsolateral rCrus I [52 -68 -26; $p_{uncorr.}$ < .001, corrected for the false discovery rate (FDR[39]) at $q$ = .05; Fig. 1a]. We then performed a general linear model (GLM) in the full sample, where we included a measure of children's ToM abilities to examine specific changes in cerebellar activations as a function of children's ToM abilities. This revealed significant clusters in the bilateral medial Crus I (rCrus I: 40 -78 -31; lCrus I: -27 -78 -32) and IV (rIV: 16 -42 -21; lIV: -8 -41 -16; $p_{uncorr.}$ < .001, FDR-corrected: $q$ = .05), which did not overlap with the Crus I clusters identified when not controlling for children's ToM abilities (Fig. 1b). These clusters thus reflected functional activation increases in the Crus I that specifically corresponded to children's increasing ToM abilities. These activation patterns remained consistent when controlling for children's age, sex, and IQ and were independent of activations related to bodily pain movie scenes, reinforcing the ToM specificity of the results (see Supplementary Figs. 1–3).

To further clarify differences between children who have and have not yet developed ToM abilities, we performed additional post-hoc analyses. We computed two separate group-level one-sample $t$-tests in children who passed the behavioral ToM task (henceforth referred to as "passers") (N = 22) and children who did not pass the task (henceforth referred to as "non-passers") (N = 19). In the group of non-passers, we identified three significant clusters in the dorsolateral rCrus I (54 -68 -30) and the bilateral IX (rIX: 3 -44 -44; lIX: -8 -44 -44; $p_{uncorr.}$ < .001, FDR-corrected: $q$ = .05; Fig. 1d). The passers additionally recruited the bilateral medial Crus 1 (rCrus I: 53 -56 -30; lCrus I: -30 -84 -30) and the rCrus II (22 -87 -36; $p_{uncorr.}$ < .001, FDR-corrected: $q$ = .05; Fig. 1c), regions which were not significantly activated in the non-passers group. These activation patterns further elucidate the group-level GLM results by showing clear differences between ToM passers and non-passers in relation to ToM processing, suggesting specific functional recruitment of the posterior cerebellum as a function of ToM emergence early in life.

We then compared these findings in children with similar analyses in two adult samples. To this aim, we computed a group-level contrast between ToM and bodily pain scenes in each of the adult samples to compare the role of the cerebellum in ToM between adults and

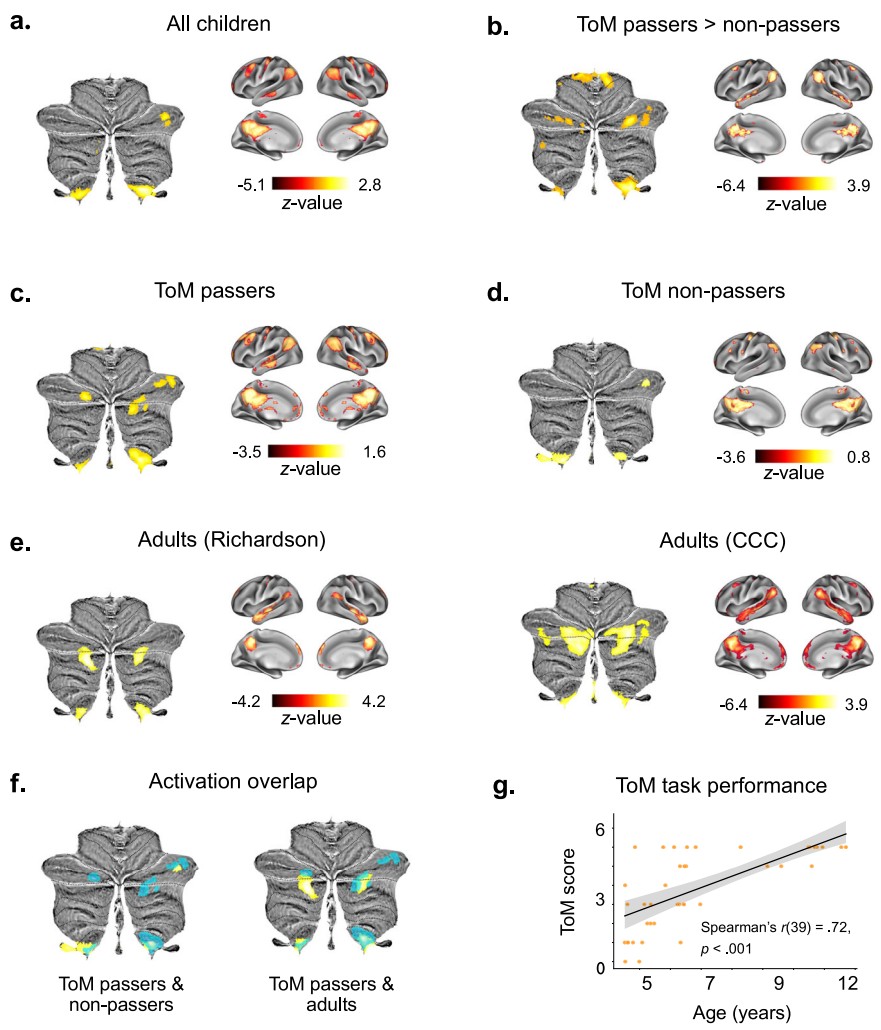

**Fig. 1 | Functional involvement of the cerebellum in ToM emergence.** Functional clusters of cerebellar and cerebral activations in the in-scanner ToM task condition (z-scored and FDR-corrected at q = .05). **a**, **c–e** One-sample t-tests showing activations for ToM vs. bodily pain movie scenes in (**a**) all children, **b** ToM passers, **c** ToM non-passers, and (**d**) adults in the Richardson et al. and Caltech Conte Center samples. **b** General linear model (GLM) of activation differences ToM passers and non-passers with children's ToM task performance (0-6) as a predictor. **f** Functional activation overlap in ToM passers and non-passers (left) and ToM passers and adults (in the Richardson et al. sample; right). Activation in ToM passers is presented in blue in each map. **g** Correlation between age and children's ToM abilities based on the out-of-scanner ToM task score (0-6). Children with a score between 4 and 6 were classified as ToM passers. ToM scores were significantly correlated with age, Spearman's $r(39) = 0.72$, $p = 9.94 \times 10^{-8}$, Cohen's $d = 2.07$, 95% CI = [0.53, 0.84]. The ribbon represents the 95% CI. All analyses were two-sided. *CCC* Caltech Conte Center.

children. In both adult samples, we consistently identified significant clusters in the bilateral medial Crus I (rCrus I: 28 -31 -81; lCrus I: -32 -82 -32), Crus II (rCrus II: 19 -88 -39; lCrus II: -18 -83 -40), and IX (rIX: 8 -53 -43; lIX: -6 -56 -42; $p_{uncorr.} < .001$, FDR-corrected: $q = .05$; Fig. 1e). Notably, the peaks of the adult clusters in the lCrus I, rCrus II and IX were similar to those in the ToM passers, suggesting an overlap of ToM regions in the developing cerebellum with mature ToM regions in the adult cerebellum.

### Functional connectivity between the posterior cerebellum and the cerebral ToM network increases as a function of ToM abilities

To examine how ToM regions in the cerebellum are functionally connected to the cerebral cortex, and how connectivity might change as a function of ToM abilities, we performed seed-to-voxel correlation analyses. In these analyses, we assessed functional connectivity between ToM regions in the cerebellum identified above (as seeds) and all voxels in the cerebral cortex. We first assessed cerebro-cerebellar connectivity from the rCrus I seed activated in all children (regardless of ToM abilities), using a one-sample t-test ($p_{uncorr.} < .001$, FDR-

corrected: $q = .05$; Fig. 1a). In the full children sample, this showed significant connections between the rCrus I and core regions of the ToM network, namely the PreC, the lTPJ, and the vmPFC, as well as connections with non-ToM network regions, namely clusters in the dorsolateral PFC, the cingulate, and the thalamus. We then assessed differences in functional connectivity between ToM passers and non-passers from this seed in rCrus I using a two-sample t-test ($p_{uncorr.} < .001$, FDR-corrected: $q = .05$). We found that ToM non-passers demonstrated greater connectivity with regions that do not belong to the cerebral ToM network (i.e., dorsolateral PFC, cingulate, thalamus; Fig. 2a, right) when compared to ToM passers.

We then focused on the connectivity of the rCrus II seed identified in ToM passers (see Fig. 1c) to better understand cerebellar connectivity with the cerebral cortex after ToM emergence. We focused on this region as it was directly related to emerging ToM abilities (based on functional activation in ToM passers described above), and due to its high overlap with ToM clusters in the adult cerebellum. The rCrus II seed demonstrated extensive connectivity with the cerebral ToM network in the ToM passers group (one-sample t-test, $p_{uncorr.} < .001$, FDR-corrected: $q = .05$). Moreover, when comparing

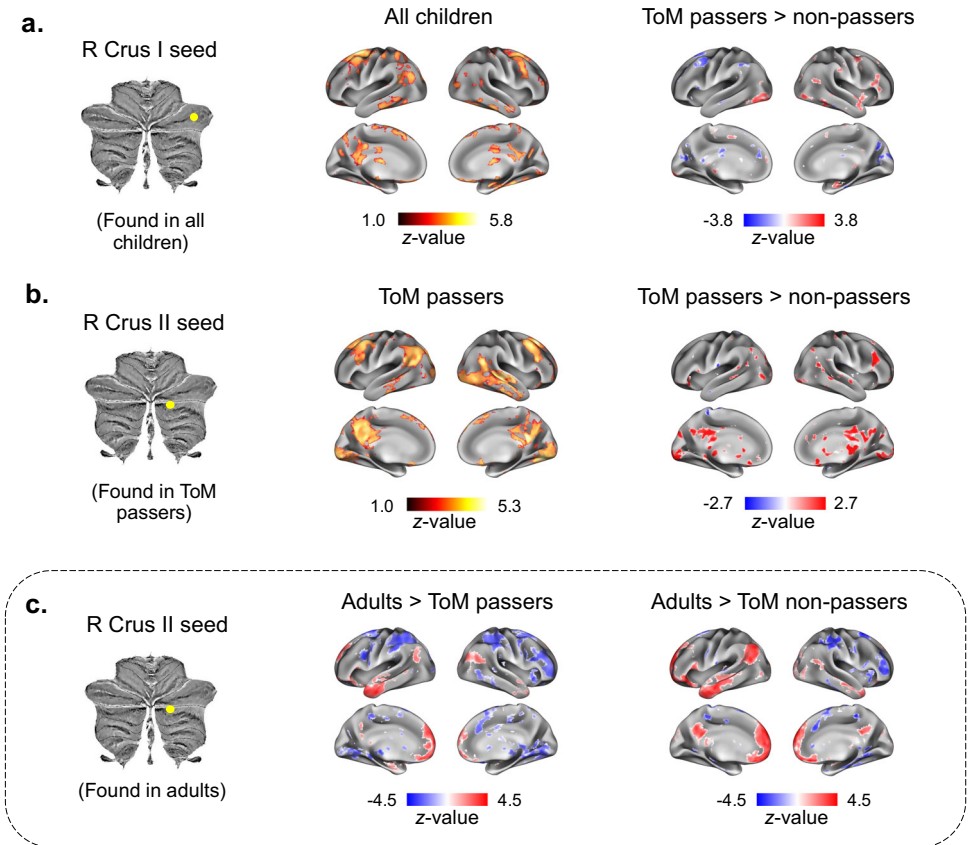

**Fig. 2 | Seed-to-voxel functional connectivity between cerebellar ToM clusters and the cerebral cortex (z-scored and FDR-corrected at $q = .05$). a** Left: one-sample $t$-test showing connectivity of right ($r$) Crus I (cluster found in all children) with the cerebral cortex in the whole sample of children. Right: two-sample $t$-test of connectivity differences between ToM passers and non-passers for rCrus I. **b** Left: one-sample $t$-test of connectivity of rCrus II (cluster found in ToM passers) with the cerebral cortex in ToM passers. Right: two-sample $t$-test of connectivity differences of rCrus II between ToM passers and non-passers. **c** Left: two-sample $t$-test of connectivity differences between ToM passers and adults (in the Richardson et al. sample) for rCrus II (cluster found in a functional atlas of ToM activations in adults[6]). Right: two-sample $t$-test of connectivity differences of rCrus II between adults and ToM non-passers. All analyses were two-sided.

connectivity for this seed in passers and non-passers, passers demonstrated greater connectivity with core regions of the ToM network, namely the PreC, rTPJ and vmPFC, than non-passers (two-sample $t$-test, $p_{uncorr.} < .001$, FDR-corrected: $q = .05$; Fig. 2b). Together, these results for the rCrus I and rCrus II seeds suggest a reorganization of cerebro-cerebellar connectivity as a function of children's ToM abilities: there was a shift from connections that include both ToM and non-ToM regions of the cerebral cortex towards more extensive and specialized connections with the ToM network as children develop ToM abilities. This is also supported by a post-hoc analysis in ToM non-passers, where the rCrus I seed defined in this group (see Fig. 1d) demonstrated connectivity primarily with non-ToM regions of the cerebral cortex as opposed to the ToM network (one-sample $t$-test, $p_{uncorr.} < .001$, FDR-corrected: $q = .05$) (Supplementary Fig. 5), further supporting the lack of specialized connectivity between the cerebellum and the ToM network before children develop ToM abilities. Importantly, the seed-to-voxel connectivity patterns for rCrus I and rCrus II were validated in resting-state data from the Baby Connectome Project (BCP)[40], which includes a comparable age range and out-of-scanner ToM assessment, suggesting the generalizability of our results and that the functional connectivity was not driven by the movie-watching task itself (see Supplementary methods and Supplementary Fig. 8).

Last, to better understand differences in ToM connectivity between the developing cerebellum and the mature cerebellum in adults, we compared cerebro-cerebellar connectivity between adults and children using two-sample $t$-tests. Here, we focused on the

Richardson et al.[26] adult sample because of the similar image acquisition parameters with the developmental sample, which allowed us to compare adults and children directly. We consistently identified greater connectivity between the rCrus II seed defined in the adult sample (see Fig. 1e) and the cerebral ToM network in adults than in children, both with and without ToM abilities (two-sample $t$-test, $p_{uncorr.} < .001$, FDR-corrected: $q = .05$; Fig. 2c). This suggests that, even though children with ToM abilities display functional connectivity between the rCrus II and the cerebral ToM network, these connections have not yet reached full maturity.

**Task-based connectivity between the posterior cerebellum and the ToM network is inverted between childhood and adulthood**
To identify the directionality of the observed connections between the cerebellum and the ToM network as a function of the in-scanner ToM task, we used DCM. We report averaged connections between pairs of ToM clusters in the cerebellum and the cerebral cortex (Fig. 3). Off-diagonal connections are expressed in units of 1/s (Hz) and represent the amount by which activation in the source cluster changes activation in the target cluster per second. For example, a positive (above-zero) connection means that activation in the source cluster increases activation in the target cluster, whereas a negative connection means that activation in the source cluster decreases activation in the target. Diagonal connections are of secondary interest and represent a cluster's self-inhibition[25]. We primarily focus on fixed connections (i.e., baseline connections that exist independently of a task) because we identified very few significant modulatory connections (i.e.,

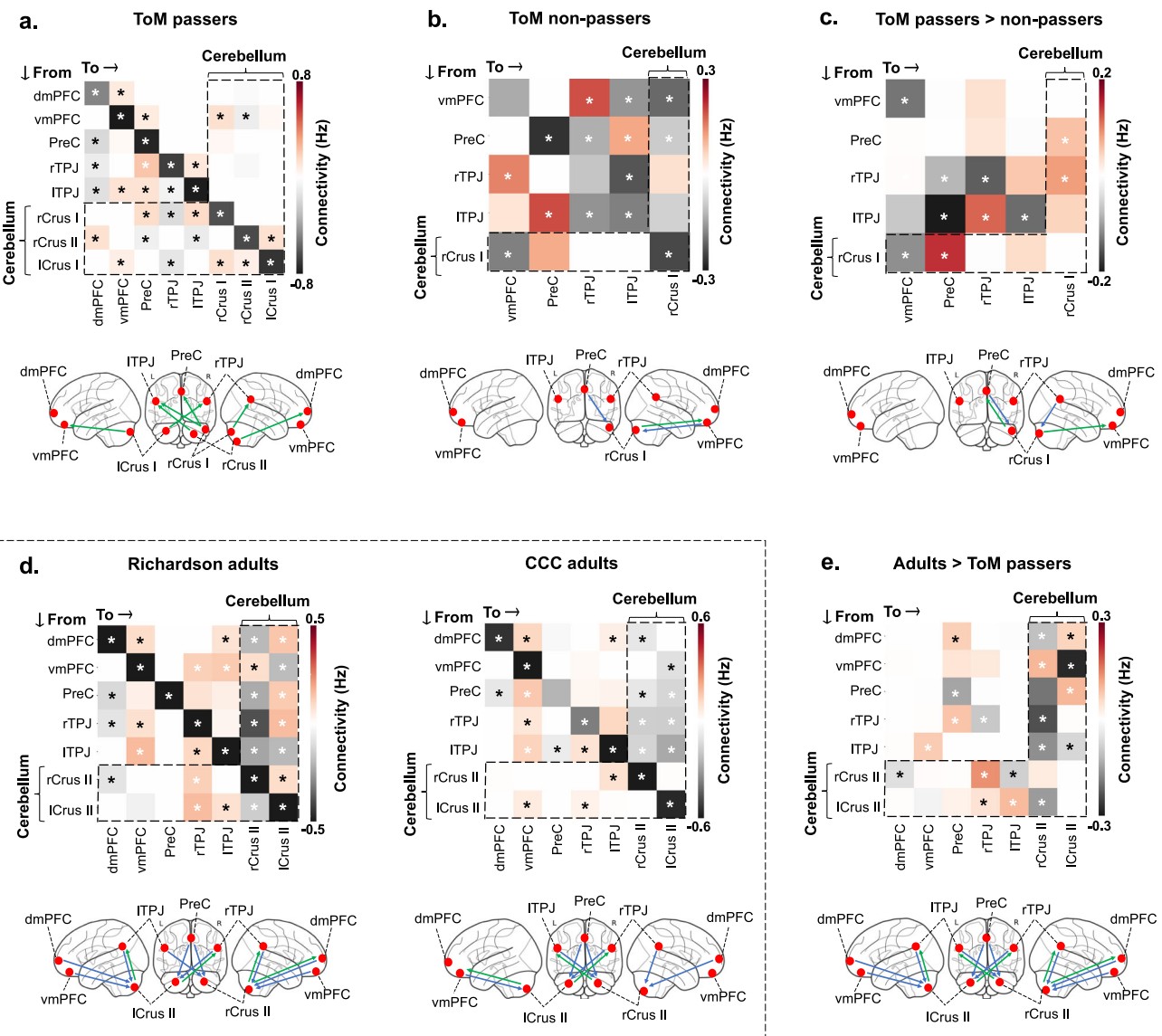

**Fig. 3 | Dynamic causal modeling (DCM) of the cerebellum and the cerebral ToM network in children and adults.** Averaged fixed (task-independent) connections ToM passers, non-passers, and adults (in units of Hz). The vertical axis represents connections that originate from a seed region and terminate in a target region (represented in the horizontal axis). **a**, **b** Fixed connections in (**a**) ToM passers and (**b**) ToM non-passers, using ToM activation clusters (identified via the in-scanner ToM task) as ROIs. **c** Connectivity differences between ToM passers and non-passers, identified by adding group type (ToM pass, ToM fail) as a covariate in the model. **d** Fixed connections in adults in the Richardson et al. and Caltech Conte Center samples using ToM ROIs identified in a functional atlas of ToM activations in adults[6]. **e** Connectivity differences between ToM passers and adults (in the Richardson et al. sample), using group type (adult, child) as a covariate. Green arrows in the glass brains represent connections from the cerebellum to the cerebral cortex. Blue arrows represent connections from the cerebral cortex to the cerebellum. Glass brains were plotted with the Nilearn Python library. * Bayesian posterior probability >.95. *CCC* Caltech Conte Center, *dmPFC* dorsomedial prefrontal cortex, *vmPFC* ventromedial prefrontal cortex, *PreC* precuneus, *r/lTPJ* right/left temporoparietal junction.

connections that reflect changes in fixed connections as a function of the in-scanner ToM movie scenes)[25]. Modulatory connections are reported in the supplementary information (Supplementary fig. 6).

To examine cerebro-cerebellar connectivity as a function of ToM abilities, we specified separate models for the groups of ToM passers and non-passers, using the ToM clusters that were significantly activated in the in-scanner movie as regions of interest (ROIs). ToM passers demonstrated significant unidirectional connections from all cerebellar ROIs to most cerebral ROIs (Bayesian posterior probability > .95, henceforth in all analyses; Fig. 3a). These connections were both contralateral (rCrus II to lTPJ and lCrus I to rTPJ) and ipsilateral (rCrus I to bilateral TPJ). There were fewer downstream cerebro-cerebellar connections from the vmPFC to right Crus I-II. Compared to ToM passers, non-passers demonstrated fewer

connections between the cerebellum and the cerebral cortex (Fig. 3b). This group showed one bidirectional connection between the rCrus I and the vmPFC, and one downstream connection from the PreC to the rCrus I.

We then directly compared the directionality of connectivity between ToM passers and non-passers in a DCM model with ROIs based on ToM movie activations for the whole developmental sample. Children's out-of-scanner ToM task performance was added as a covariate in the model. This showed significant differences between fixed connections in ToM passers and non-passers for the rCrus I seed (Fig. 3c). Specifically, passers demonstrated stronger bidirectional connections between the cerebellar rCrus I and the PreC than non-passers, as well as stronger connections from the lTPJ to the rCrus I, and from the rCrus I to the vmPFC.

To compare cerebro-cerebellar connectivity in children with that one observed in adults, we conducted additional DCM analyses in the two adult samples. Connectivity patterns were consistent across both adult samples (Fig. 3d). We observed significant fixed connections from most of the cerebral ROIs to the cerebellar ROIs in both datasets. In contrast to the developmental sample, upstream fixed connections from the cerebellum to the cerebral ROIs were less frequent (see Fig. 3d). Modulatory connections followed a similar pattern, in which the majority of connections modulated by the ToM task were directed towards the cerebellar ROIs from the cerebral ToM network (Supplementary Fig. 6a).

To directly compare cerebro-cerebellar ToM connectivity in children and adults, we specified an additional model with adults (Richardson et al. sample) and ToM passers, where we included group (adult or child) as a covariate (Fig. 3e). Adults demonstrated higher connectivity than ToM passers from most cerebral ToM regions towards cerebellar ToM regions, especially for the rCrus II. Upstream connections from the cerebellum to the cerebral cortex were less frequent (see Fig. 3E). A similar pattern was observed in modulatory connections, where adults demonstrated higher connectivity than ToM passers from the bilateral Crus II to the vmPFC (Supplementary fig. 6d). Overall, adults seemed to follow an inverse pattern compared to that of children with ToM abilities. That is, in adults, fixed and modulatory connections primarily originated from the cerebral ToM network and terminated in cerebellar ToM regions, whereas in children, connections originated from cerebellar ToM regions and terminated in the cerebral ToM network.

Notably, in all analyses, the polarity (i.e., positive or negative sign) of cerebro-cerebellar connections did not reveal specific patterns. Connections to and from the cerebellum were either positive or negative, meaning that there was no clear pattern in the extent to which activation in ToM regions in the cerebellum increased or decreased activation in the cerebral cortex, and vice versa. Similar results have been obtained in recent publications on task-driven cerebro-cerebellar connectivity in the context of social cognition[22,23].

## Discussion

In the present study, we examined the functional involvement of the cerebellum in the development of ToM early in life. An in-scanner movie with ToM scenes revealed significant involvement of the posterior cerebellum, specifically Crus I-II, in ToM in young children. These activations were associated with the emergence of important ToM abilities: children who passed an out-of-scanner false-belief task demonstrated activation clusters in Crus I-II similar to clusters observed in adults, which were not significantly activated in children who did not pass the false-belief task. These regions also showed increased functional connections with the ToM network in the cerebral cortex as a function of children's ToM abilities. Moreover, task-based connections from the cerebellum to the cerebral cortex were more prominent in ToM task passers, whereas in adults, downstream connections from the cerebral cortex to the cerebellum were more prominent.

Our results provide evidence for the involvement of the cerebellum in the emergence of ToM abilities early in life. We found that functional changes in cerebellar Crus I-II and their connectivity to the cortex were associated with the emergence of false-belief reasoning in the critical period between 3 and 5 years. This highlights the importance of the cerebellum, not only in adult social cognition[6–8,21], but in the development of ToM in early childhood. Our finding that connections from the cerebellum to the cerebral cortex were important in ToM passers is in line with a prominent theory that the cerebellum plays a major role in early predictive social processing, by storing internal models of social sequences which can be fed forward to the cerebral cortex to allow predictions about social interactions[41–43]. In accordance with this theory, our findings suggest a significant

association of early-life changes in the cerebellum with ToM development, insofar as the cerebellum stores information that supports the subsequent prediction of mental states. Our results also align with evidence suggesting that the sociocognitive deficits of ASD are associated with functional and structural abnormalities of the cerebellum[12,34,44,45]. Our observation of the role of the Crus I-II in ToM development may function as a framework of cerebellar regions and cerebro-cerebellar connections crucial for ToM in typically developing children, generating hypotheses for targeted studies in these regions in pediatric populations (e.g., children with ASD).

By assessing functional connectivity of cerebellar ToM regions with the cerebral cortex, we found evidence of changes in connectivity profiles as a function of children's emerging ToM abilities. As children's ToM abilities emerged, there was a switch from connections with a broader, non-ToM-specific network of cerebral cortex regions to extensive and almost exclusive connections with the cerebral ToM network. Specifically, the rCrus I, which was a cluster activated during ToM scenes in all children, demonstrated connections with core regions of the ToM network, as well as the dorsolateral PFC, cingulate, and thalamus. However, the rCrus II, a ToM cluster identified only in ToM passers, was predominantly connected with the ToM network, indicating its specificity for ToM. In contrast, ToM non-passers demonstrated dominant connections of the cerebellum with non-ToM regions in the cerebral cortex (dorsolateral PFC, cingulate, and thalamus). Together, these patterns demonstrate greater functional specialization of cerebro-cerebellar connectivity as a function of ToM abilities in early childhood. Younger children seemed to recruit regions primarily associated with general executive control, such as memory, prevention of interference, and attention, when processing ToM scenes in a movie[46,47]. In turn, when children's false-belief abilities emerge, their cerebellum starts to exchange information with the cerebral ToM network. While the connections between the rCrus II and the ToM network observed in ToM passers were similar to those of adults, these connections became stronger in adulthood, suggesting that the cortico-cerebellar ToM network continues to mature in later childhood.

We additionally explored task-based connectivity between ToM regions in the cerebellum and the cerebral cortex using DCM. Consistent with functional connectivity, ToM passers demonstrated more extensive connectivity than non-passers between all cerebellar ToM regions and the cerebral ToM network. Our results further suggested inverse patterns of cerebro-cerebellar connectivity between childhood and adulthood, with dominantly upstream connections from the cerebellum to the cerebral ToM in childhood but dominantly downstream connections in adults. This was observed in fixed (task-independent) connections in adults and children as well as in modulatory (task-dependent) connections in adults. Our findings provide empirical support for recent conceptualizations of a developmental gradient of cerebro-cerebellar connectivity[10]. In this view, upstream connections from the cerebellum to the cerebrum are proposed to be particularly important early in life as they aid the initial construction of social mental models, whereas downstream connections from the cerebrum to the cerebellum allow the fluid utilization of these models and are thus more prominent in adulthood. This is in line with evidence showing that early-life lesions in the posterior cerebellum lead to persistent ToM deficits, whereas lesions in adulthood are less dramatic and often impact the fluidity of ToM reasoning (see ref. 10 for a review).

Nevertheless, it should be noted that the directionality of cerebro-cerebellar connections in DCM should be interpreted with caution as the output of the cerebellar Purkinje cells is primarily inhibitory, so that their activation cannot be modeled well based on blood-oxygen-level-dependent (BOLD) signal changes[48–50]. However, our finding of bidirectional cerebro-cerebellar connections converges with research showing bidirectional white matter loops between the cerebellum and

the cerebral cortex[18,51–54], and specifically the cerebellar Crus I-II and the cerebral ToM network[7]. Moreover, the asymmetry in cerebro-cerebellar connectivity between childhood and adulthood is further corroborated by clinical evidence that early cerebellar lesions severely impact social-cognitive development, whereas later lesions do not, which is in line with our findings suggests a crucial role of the cerebellum, particularly early in ToM development[10]. Our findings of greater reliance on feedforward connections from the cerebellum to the ToM network during development offer a possible explanation for the developmental asymmetry of the impact of cerebellar lesions observed in clinical populations. Future research should examine the directionality of cerebro-cerebellar connections with methods allowing for the assessment of the causal relation between cerebral and cerebellar activation in ToM more directly. For example, transcranial magnetic stimulation (TMS) studies could assess whether inhibiting the activity of ToM regions in the cerebellum or the cerebral cortex differentially affects ToM (predicting that activity inhibition in the ToM network will impair ToM processing more than the cerebellum). However, it should be noted that such methods can only be employed in adults and not in early childhood.

A further limitation is our developmental sample size. Given that most participants in the original dataset had incomplete cerebellar coverage, we limited our analyses to 41 children. Given the modest sample size in this analysis, future studies should further investigate the role of the cerebellum in ToM emergence in larger samples. Additionally, the use of both resting-state and movie-induced brain states in our validation analysis may present interpretational challenges, as these conditions engage distinct cognitive processes. While we used both states to investigate functional connectivity patterns, we acknowledge that they are not directly equivalent and may tap into different aspects of neural function. Nevertheless, the consistency of findings across the movie-based primary dataset and the resting-state validation dataset strengthens our confidence in the generalizability of the results.

An additional consideration is possible anatomical differences in the context of adult-child comparisons. Following Richardson et al.'s[26] original study, we normalized child and adult brains to the same standard template. As in the original study, this was motivated by similar procedures extending to children under 7 years[26,55]. Nevertheless, we acknowledge the existence of shape and size differences between the developing and the adult brain, which could have influenced our results.

Our results primarily reflect associations between cerebellar function and individual differences in the development of ToM, without making claims about the prediction of ToM task performance based on cerebellar activation across different developmental or clinical contexts. Although our sample size does not currently support predictive inferences, we believe our findings provide a valuable framework for future research by identifying cerebellar regions directly associated with ToM development early in life, thereby opening exciting opportunities for predictive modeling.

Our developmental sample was primarily composed of younger children (mean age: ~6 years), with fewer observations in the 8-12-year age range ($N = 10$). The observed differences between childhood and adulthood raise the exciting question of how cerebellar activations and cerebro-cerebellar connectivity in the context of ToM change through later childhood and adolescence until adulthood. For example, future studies could investigate cerebro-cerebellar connectivity in ToM at different stages of development to see when and how the directionality of cerebro-cerebellar connectivity switches from an upstream model in childhood to a downstream model in adulthood.

Further, it would be interesting to explore the function of cerebellar activations in younger children and infants. Even though verbal false-belief ToM reasoning only emerges between 3-5 years of age, in non-verbal tasks, infants younger than 2 years already seem to consider others' mental states[56,57]. These infant abilities have been shown to rely on cortical regions that are independent of the cerebral ToM network[31]. In the present study, we found that children who failed in verbal false-belief tasks relied more on different cerebellar clusters than children who succeeded, namely the dorsolateral rCrus I (as opposed to medial Crus I-II), which was functionally connected to the dorsolateral PFC, cingulate, and thalamus. Future studies should assess the extent to which these regions relate to the non-verbal abilities observed in infants.

Overall, the current study found that functional changes in cerebellar Crus I-II and their connectivity to the cerebral ToM network are associated with the emergence of ToM abilities in early childhood. Cerebro-cerebellar connectivity in children and adults seems to follow inverse directions, where connections from ToM regions of the cerebellum to the ToM network are more prominent in childhood and connections from the ToM network to ToM regions of the cerebellum are more prominent in adulthood. This indicates that the cerebellum may be of particular importance to ToM early in development and may play a more supportive role in adulthood. Together, our results suggest a crucial involvement of the cerebellum in typical ToM development, a milestone of human social cognition, with important implications for the onset of social-cognitive disorders early in life.

## Methods

### Preregistration
As no previous research on our research question and analytical approach existed, we used the adult data to inform our methods and hypotheses, and then preregistered our developmental analyses prior to analyzing the developmental data (https://aspredicted.org/3R4_JTY).

### Participants
We leveraged openly accessible adult and developmental data from two previous studies, which investigated social cognition in adults and typically developing children[26,36]. The Richardson et al.[26] dataset includes data of 122 children who watched an in-scanner ToM movie. Children additionally performed an out-of-scanner assessment of ToM abilities, namely a battery of behavioral false-belief (ToM) tasks (see "Behavioral task battery"). Children were excluded from the original study if they did not complete the behavioral and neuroimaging components, if they moved excessively in the scanner (see "Preprocessing"), or if they demonstrated language delays. Since these data were not collected specifically for the cerebellum, we further excluded children with insufficient cerebellar coverage in BOLD scans after visual inspection ($N = 71$), with our final sample consisting of 41 children (age range: 3-12 years; age: $M$ ($SD$) = 5.91 (2.29) years; 25 female). No statistical method was used to predetermine sample size. For our analyses, we binarized the ToM task scores of all children into ToM passers and non-passers (see "Behavioral task battery"), yielding subsamples of 22 ToM passers (age: $M$ ($SD$) = 7.20 (2.40) years; 15 female) and 19 non-passers (age: $M$ ($SD$) = 4.42 (0.78) years; 10 female).

We further used the adult sample from the Richardson et al.[26] dataset. For validation purposes, we also examined adults from the Caltech Conte Center dataset[36], a recent multimodal resource for exploring social cognition. These datasets contain movie-watching fMRI data of 30 and 58 adults, respectively. We excluded 8 participants from the Richardson et al. dataset for not having sufficient cerebellar coverage. We further excluded 2 participants with poor fMRI data quality from the Caltech Conte Center dataset, as indexed by three independent raters' visual examination of the preprocessed data[36]. Thus, our final samples consisted of 22 adults (age: $M$ ($SD$) = 24.91 (5.33) years; 14 female) in the Richardson et al. dataset, and 56 adults (age range: 18-48 years; age: $M$ ($SD$) = 30.45 (7.27) years; 22 female) in the Caltech Conte Center dataset. Across all samples, data were acquired in accordance with the ethical procedures of the

participating institutions' review boards and with participants' informed consent (or the consent of the legal guardian, in case the participants were children). The corresponding authors' institution did not require additional ethical approval for secondary analysis of open data.

## Behavioral task battery

All children performed an out-of-scanner verbal ToM assessment[55]. In this task, children listened to an experimenter narrate a story and then answered prediction and explanation questions about the mental states of the story characters (e.g., their emotions, beliefs, or moral blame-worthiness). Here, we used the questions targeting children's understanding of false beliefs, as false belief understanding is considered the critical test of ToM abilities[27]. There were six canonical ToM questions, which involved unexpected changes in an object's location or content[28,58]. For example, children were asked where an agent with a false belief about an object's location would look for the object, and why. Based on children's performance in these questions, a composite false-belief score was created for each child. Children who answered less than half of the questions correctly (0-3) were considered ToM non-passers, and children who answered more than half of the questions correctly (4-6) were considered ToM passers. The composite score demonstrated acceptable reliability in the original study (Cronbach's $\alpha = 0.71$). The exact task narrative and instructions can be found on the open science framework (OSF) (https://osf.io/ G5ZPV/). Children performed additional out-of-scanner tests of cognitive abilities, namely a measure of non-verbal IQ (under 5 years: WPPSI block design[59]; over 5 years: non-verbal KBIT-II[60]).

## In-scanner ToM movie

Participants across all samples watched a 5.6 min animated video (Pixar's "Partly Cloudy"). The movie began after 10 s of rest. A short description of the plot can be found online (https://www.pixar.com/ partly-cloudy#partly-cloudy-1). The movie includes scenes that depict bodily pain (e.g., an alligator biting the main character) and scenes that trigger ToM (e.g., the main character revealing their intention). Seven ToM and nine pain event timepoints (ToM: 60 s total, $M$ ($SD$) length: 8.6 (4.6) s, pain: 66 s total, $M$ ($SD$) length: 7.3 (4.4) s were defined in the original Richardson et al. (2018) study. These scenes have been validated as activating regions of ToM and pain processing in adults and children[26,61].

## fMRI data acquisition

fMRI acquisition parameters for structural (T1-weighted) and functional MRI data are described in detail elsewhere[26,36]. Briefly, in Richardson et al.[26], whole-brain structural and functional images were acquired on a 3 T Siemens Tim Trio scanner. Children under five used a custom 32-channel phased-array head coil to accommodate their head size, while older participants used a standard Siemens 32-channel head coil. T1-weighted images were acquired in 176 interleaved sagittal slices with 1 mm isotropic voxels [GRAPPA parallel imaging, acceleration factor of 3; FOV: 256 mm (adult coil); 192 mm (children coils)]. Functional data were collected with a gradient-echo EPI sequence in 32 interleaved near-axial whole-brain slices aligned with the anterior/ posterior commissure (EPI factor: 64; TR: 2 s, TE: 30 ms, flip angle: 90°). Functional data were upsampled to 2 mm isotropic voxels. Children under five completed two functional runs, and older participants one run. 168 volumes were acquired in each run, and only the first run was analyzed for all participants. To ensure smooth participation in the scanning, children completed a mock scan to get acquainted with the scanner. While in the scanner, an experimenter stood by their feet to monitor whether children paid attention during the functional task.

Structural and functional data in the Caltech Conte Center dataset[36] were acquired with a 3 T Siemens Prisma.Fit scanner with a 32-channel head receive array coil. The Caltech Conte Center dataset

contains multiple acquisition protocols reflecting changes in MRI data collection throughout the years of the project. Here, we focused on protocol version 2.2, which was the one used for the in-scanner ToM movie. T1-weighted structural data were acquired with 0.9 mm isotropic voxels (multi-echo MEMP-RAGE pulse sequence; acceleration factor of 2; flip angle: 7°; water excite fat suppression). Functional data were collected with a multi-band 2.5 mm isotropic T2*-weighted EPI sequence (EPI echo spacing: 0.49 ms; TR: 0.7 s, TE: 30 ms, flip angle: 53°). Movie-watching was completed in a single run, where 476 volumes were collected.

## fMRI data analysis

**Preprocessing.** We used preprocessed structural and functional data, which had undergone robust motion artifact removal and quality control in the original datasets. Briefly, functional data in Richardson et al.'s[26] study were registered to the first image of the run, co-registered to each participant's anatomical image, and the anatomical image was normalized to the ICBM/MNI 152 2009c Nonlinear Asymmetric space (MNI152NLin2009cAsym) template[62]. Data were then smoothed with a 5 mm Gaussian kernel. Denoising involved motion artifact detection via the ART toolbox (https://www.nitrc.org/projects/ artifact_detect/), where artifacts were defined as timepoints displaying >2 mm composite motion relative to a previous timepoint or timepoints where the global signal was over 3 $SDs$ relative to all participants' mean global signal. Additionally, five principal component analysis (PCA)-based noise regressors were generated via CompCor within subject-specific white matter masks[63]. ToM passers and non-passers were balanced in terms of motion artifact timepoints. A Welch's $t$-test showed that the number of artifact timepoints did not differ significantly between ToM passers ($M = 10.18$; $SD = 10.38$) and non-passers ($M = 43.33$; $SD = 12.59$), $t(39) = 1.21$, $p = 0.27$, Cohen's $d = -2.88$, 95% CI $= [-40.51, -25.79]$.

Functional and structural data in the original Caltech Conte Center dataset[36] were processed with a multi-step pipeline, including fMRIPrep 20.2.1[64]. In summary, fMRIPrep processing involved registration of functional images to a reference volume and co-registration to each participant's anatomical image, slice-timing correction and resampling into the MNI152NLin2009cAsym standard space. Here, we additionally applied a 5 mm Gaussian kernel to the preprocessed data for consistency with the Richardson et al.[26] data. For the same reason, we also focused on PCA-derived CompCor regressors defined in the original dataset for denoising the functional data: six CompCor regressors were calculated within the intersection of the subcortical mask, the cerebrospinal fluid and the white matter mask for each participant.

**Cerebellar isolation and spatial normalization.** As opposed to the original studies that had only analyzed the cerebrum, here we focused on the cerebellum. The cerebellum was isolated and normalized to the spatially unbiased infra-tentorial template (SUIT)[65] using the SUIT toolbox (v.3.5) (https://github.com/jdiedrichsen/suit/releases/tag/3.5), implemented in SPM12[66] within MATLAB R2022b[67]. Normalization of the cerebellum to a spatially unbiased template has been shown to greatly improve the overlap of individual cerebellar fissures and deep cerebellar nuclei compared to normalization to the standard ICBM152 MNI space[21,65,68,69]. As a first step, individual participants' cerebella were extracted and segmented into gray and white matter, and a cerebellar isolation mask was constructed. This mask was visually inspected and hand-corrected for each subject using MRIcron (v.1.0.2019) (https:// www.nitrc.org/projects/mricron)[70], to ensure that no voxels from the inferior temporal and occipital cerebral cortex were included. The individual gray and white matter probabilistic maps were then normalized into the SUIT space using the diffeomorphic anatomical registration through the exponentiated Lie algebra algorithm[71]. In this approach, the cerebellum is deformed in order to simultaneously fit

the cerebellar gray and white probability maps onto the SUIT atlas template. This nonlinear deformation was applied to individual participants' anatomical and functional data, specifically the first-level contrast images (see "Contrast analyses"). All images were masked with the cerebellar isolation mask to prevent activation influences from the occipital cortex. All data were displayed on the cerebellar flatmap, a surface-based reconstruction of the cerebellar cortex which allows the spatial extent of task-based activations in the cerebellum to be fully visualized. The flatmap was only intended for visualization purposes, given that it is not an accurate representation of cerebellar folding[6,21].

**Contrast analyses.** We used the Nilearn library (version 0.10.1)[72] in Python (version 3.9.7) to identify ToM activations in the cerebellum and the cerebral cortex. We conducted separate analyses for the entire sample of adults in the Richardson et al.[26] dataset, adults in the Caltech Conte Center[36] dataset, and children in the Richardson et al. dataset. First, whole-brain ToM activations were identified via the ToM scenes > bodily pain scenes contrast on an individual-subject level ($p < 0.001$, uncorrected). The onset and duration of each ToM and pain event were specified and convolved with the standard hemodynamic response function. As in Richardson et al., across all samples, a number of artifact timepoints and PCA-based CompCor noise regressors were included as parameters of no interest to account for motion artifacts. The level-one statistical maps were then passed on to separate group-level one-sample $t$-tests (two-sided; $p < 0.001$, uncorrected) for children and each of the adult samples to test for the significance of ToM activations in the ToM > pain contrast against zero. We controlled for false positive activations via FDR[39] on a .05 level. Group-level analyses were conducted separately for the cerebellum and the cerebral cortex (see "Cerebellar isolation and spatial normalization").

We performed additional group-level analyses in the developmental sample to test for differences in ToM activations depending on children's ToM abilities. To this aim, we first conducted a level-two general linear model (GLM) with children's ToM score (0-6) as a ratio predictor to investigate differences in cerebellar activations as a function of ToM abilities. To better understand differences between children who passed and failed the ToM assessment, we performed separate two-sided one-sample $t$-tests for ToM passers and non-passers (see "Participants"). Lastly, to test for ToM specificity in our results, irrespective of children's general executive functions, we performed additional GLM analyses in our entire sample of children, where we used children's IQ scores, age, and sex as additional predictors (see "Supplementary methods" and Supplementary Figs. 1–2). As described above, all analyses were conducted separately for the cerebellum and the cerebral cortex, with an initial alpha level of .001 (uncorrected), and an FDR threshold of $q = .05$.

**Seed-to-voxel correlations.** We performed seed-to-voxel correlation analyses between ToM seeds in the cerebellum and voxels of the cerebral cortex using Nilearn 0.10.1 in Python 3.9.7. In adults, ToM ROIs in the cerebellum were identified based on local maxima in the ToM task contrast defined in seminal work by King and colleagues[6], after setting the ToM contrast map threshold to the top 10% of activations with a cluster extent threshold ($k$) of 5. This led to the rCrus II (26 -80 -39) and lCrus II (-26 -78 -38) as our cerebellar ToM ROIs. These clusters highly overlap with regions identified in other studies examining the role of the cerebellum in adult ToM[24]. We extracted the BOLD signal within 5 mm spheres centered around these MNI coordinates, while controlling for physiological noise regressors. In children, we defined ROIs with a different approach, given that there are no studies that have identified the ToM network of the developing cerebellum. ROIs were centered around the MNI coordinates of local maxima derived from the sample's group-level results of the ToM > pain contrast ($p < .001$ uncorrected; FDR threshold: $q = .05$; $k = 5$). We defined multiple sets of ToM ROIs corresponding to local maxima in our level-two contrast

maps: i) a set for the whole developmental sample ($N = 41$); ii) a set for ToM passers ($N = 22$); iii) a set for ToM non-passers ($N = 19$) (see "Results" and Supplementary tables 1–3 for the MNI coordinates of the ROIs). The anatomical locations of the local maxima were verified with the SUIT probabilistic atlas[65].

We correlated the BOLD signal within each of the cerebellar ToM ROIs and every voxel in the cerebral cortex for every participant. Single-subject seed-to-voxel correlation matrices (corrected for nuisance regressors, as above) were passed on to a group-level two-sided one-sample $t$-test to identify significant seed-to-voxel functional connections against zero ($p < .001$ uncorrected; FDR threshold: $q = .05$). We conducted separate one-sample $t$-tests for each adult sample, the whole developmental sample, ToM passers, and ToM non-passers. Lastly, we directly compared functional connectivity between ToM passers and ToM non-passers, between adults and ToM passers, as well as adults and ToM non-passers with separate level-two independent $t$-tests (two-sided; $p < .001$ uncorrected; FDR threshold: $q = .05$). For these comparisons, we first repeated the single-subject seed-to-voxel correlations with different sets of ROIs, namely: i) the ROIs identified in the whole developmental sample and the ROIs identified in ToM passers for GLM analyses; ii) the adult ROIs for comparisons between adult and children (especially since ToM passer ROIs closely resembled adult ROIs). In adult-children comparisons, we only focused on adults from the Richardson et al. study, given the similar data acquisition parameters to children's data. Connectivity differences as a function of ToM abilities in development were additionally tested, accounting for children's age and sex (see Supplementary methods and Supplementary Fig. 4).

**Dynamic causal modeling.** For the DCM analyses, we followed the approach outlined in previous studies that tested the task-based or effective connectivity between the cerebellum and the ToM network in the cerebral cortex[22,23,25,43]. All analyses were conducted in SPM12 within MATLAB R2021a. In adults, cerebral cortex ROIs for the DCM analyses were centered around MNI coordinates based on previous meta-analyses on the ToM network[73,74], and included the bilateral TPJ (±50 -55 25), the dmPFC (0 50 35), the vmPFC (0 50 5), and the PreC (0 -60 40). In the cerebellum, we identified ROIs using the King et al.[6] ToM cerebellar activation map (see "Seed-to-voxel correlations"). Cerebral and cerebellar ROIs were created for the ToM > pain contrast by setting a whole-brain threshold at $p < .05$ (uncorrected), while controlling for nuisance regressors. In the cerebral cortex, we extracted the eigenvariate time series within an 8 mm sphere centered around the nearest local maximum within 15 mm of the corresponding ROI centers described above. In the cerebellum, we repeated this process but created 5 mm ROIs within 11 mm of the corresponding cerebellar ROI, to account for the smaller size of the cerebellum[25,43]. All ROI voxels were adjusted for the effects of interest (i.e., the ToM and pain conditions in the ToM movie). We defined subject-specific ROIs using a progressively tolerant threshold to ensure that activations from all ROIs were included in the DCM models[25,43]. That is, if no voxel survived using a $p < .05$ threshold, we repeated this process with uncorrected thresholds of $p < .10$ and $p < 1.00$ (in which case the ROIs were centered around the group-based coordinates detailed above)[75]. This way, we increased the robustness of our DCM results by including activations of all participants for all given ROIs[25,43,75]. The thresholds used for participants in each ROI are detailed in Supplementary table 4.

In children, we identified the centers of cerebral and cerebellar ROIs using the same approach we followed for seed-to-voxel correlations, namely based on local maxima in the group-level ToM activation map. The anatomical locations of the local maxima were verified with a meta-analytic resource (NeuroSynth)[76] for the cerebral cortex and the SUIT probabilistic atlas[65] for the cerebellum. Note that defining ROIs based on group-level results is not circular in DCM analyses: DCM does not test whether a certain region demonstrated an experimental effect

but compares different hypotheses about the neuronal mechanisms that underlie experimental effects in a certain region[38]. For example, DCM evaluates whether an experimentally driven activation identified in a conventional statistical parametric mapping (SPM) model was due to changes in a region's self-inhibition or due to a modulation of one or several afferent connections of this region by the experimental condition. We then defined individually tailored ROIs for the whole sample of children, for ToM passers and for ToM non-passers using the same method we used in adults. The ROIs located in the bilateral IX were subsequently dropped from the main analyses since not all individual children had voxels that survived within these ROIs, even after setting the threshold to $p < 1.00$. Specifically, 11 out of 41 children had no surviving voxels in the rIX and lIX (5 passers in rIX, and 4 non-passers in lIX). We still report DCM results for the subsample of children with activations in the cerebellar IX in the supplementary information (Supplementary fig. 7).

DCM models were estimated based on the procedures outlined in refs. [77,78] (see also: https://en.wikibooks.org/wiki/User:Peterz/sandbox). We performed separate analyses for adults in the Richardson et al. and Caltech Conte Center datasets, and separate analyses for children. In the latter case, we first constructed separate models for the whole sample of children, in which we either controlled for children's ToM score by adding it as a covariate in the models, or did not. Children in the covariate model were classified as ToM passers or non-passers, and these categories were dummy-coded (+1 for passers and -1 for non-passers) and centered around the mean to reflect the average connectivity between parameters across the groups. We also constructed additional models in which we only included ToM passers ($N = 22$) or ToM non-passers ($N = 19$) to better understand how different regions of the cerebellar or cerebral ToM networks interact in children with varying ToM abilities. Lastly, we built a model in which we collapsed adults and ToM passers, in order to directly compare cerebro-cerebellar connectivity in ToM processing in childhood and adulthood. As in seed-to-voxel correlations, we only focused on adults from Richardson et al. We defined our model based on adult ToM ROIs in the cerebellum and the cerebral cortex for adults and ToM passers, since cerebellar ToM activations of ToM passers closely resembled activations in adults and added participant group (adult or child) as a covariate in the model.

In all analyses, we specified a full bilinear deterministic DCM (SPM function *spm_dcm_fit*) on a single-subject level, without centering around the mean[37], which included: i) all forward and backward fixed (or endogenous) connections between the ROIs; ii) all modulatory connections which reflected connectivity changes due to the ToM condition; iii) direct input parameters which reflected input driving connectivity between ROIs in both the ToM and pain conditions, by combining onsets of both conditions in a single vector[79]. Specifying a full model allowed us to freely estimate all connectivity parameters in all directions.

On the group level, we constructed a parametric empirical Bayes (PEB) model over all connectivity parameters (SPM function *spm_dcm_peb*), which makes it possible to estimate the group-average effective connectivity while taking within-participant variability into account. A PEB was also chosen because it allowed us to control for individual differences between participants (e.g., passing or failing the ToM test) by including them as a covariate in some of our models. Last, we used Bayesian model reduction (SPM function *spm_dcm_peb_bmc*) to automatically remove connectivity parameters that did not contribute to the model evidence from the group-level PEB model, by performing a greedy search, as recommended by ref. [77]. In this approach, connectivity parameters are pruned from the full PEB model until model evidence starts to decrease, thus maintaining only the most relevant models. The winning model is then determined by performing Bayesian model averaging of the 256 most relevant models, which is then used for group inferences[75]. Connectivity parameters whose posterior probability was over $p > .95$ were considered significant. Note that following a Bayesian approach in the single-subject DCM and group-level PEB estimations circumvents the multiple comparisons problem[37].

## Reporting summary

Further information on research design is available in the Nature Portfolio Reporting Summary linked to this article.

## Data availability

This study includes a re-analysis of existing neuroimaging data. All raw and preprocessed data for this study are available on OpenNeuro (Richardson et al.[26]: https://openneuro.org/datasets/ds000228/versions/1.1.0; Caltech Conte Center[36]: https://openneuro.org/datasets/ds003798/versions/1.0.5).

## Code availability

Scripts for all analysis pipelines are available on GitHub (ref. [80]; https://doi.org/10.5281/zenodo.15212745).

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

## Acknowledgements

We thank Jörn Diedrichsen for insightful comments on data analysis and results interpretation. This work is supported by the Max Planck Society. A.M. was also funded by the German Academic Scholarship Foundation (Studienstiftung des deutschen Volkes). S.L.V. was also funded in part by Helmholtz Association's Initiative and Networking Fund under the Helmholtz International Lab grant agreement InterLabs-0015, and the Canada First Research Excellence Fund (CFREF Competition 2, 2015-2016) awarded to the Healthy Brains, Healthy Lives initiative at McGill University, through the Helmholtz International BigBrain Analytics and Learning Laboratory (HIBALL). S.L.V. was furthermore supported by the Jacobs Foundation Research Fellowship and the Hector Foundation Research Development Award. C.G.W. was supported by the ERC Starting Grant (REPRESENT 101117806).

## Author contributions

A.M., S.L.V. and C.G.W. conceptualized and designed the study. A.M. analyzed the data with input from S.L.V., C.G.W. and F.V.O. A.M. wrote the original draft and revised the manuscript. All authors edited and reviewed the final manuscript.

## Funding

## Competing interests

The authors declare no competing interests.
