## [Transparent Peer Review file · Nature Communications]

Functional recruitment and connectivity of the cerebellum is associated with the emergence of Theory of Mind in early childhood

Corresponding Author: Ms Aikaterina Manoli

Version 0:

Reviewer comments:

Reviewer #1

(Remarks to the Author)

Manoli et al analyse publicly available fMRI data from children and adults watching a short animated film, to examine a previously untested question: the link between theory of mind and cerebellar activity in children. They find that cerebellar regions in CRUS-I/II are active during 'mentalistic' scenes in the movie, in children as in adults, and that both the location of the activity, and correlations with cortical regions, are more adult like in children who (according to independent behavioral measures) have more sophisticated theory of mind abilities. I believe this paper is a useful contribution to the literature. There has been increasing attention to and evidence for the cognitive functions of the cerebellum, and these results fit with that growing consensus.

I have a few major concerns that must be addressed before considering publication.

First, although many of the analyses compare neural activity ToM "passers" and "failers", I cannot tell whether all of those analyses control for chronological age and fMRI data quality — both which are correlated with ToM task performance and could confound measures of neural differences.

Second, the functional correlation analyses were conducted during the same movie as the contrast based analyses. I am concerned that this is a major confound: it is impossible to differentiate correlated spontaneous fluctuations (nominally evidence of 'functional connectivity' between regions) and correlations that arise from the movie stimulus itself activating regions at the same time. So, the finding of different movie-evoked activity, and different "functional correlations" may be just two ways of describing the same data. This hypothesis would be much better tested in resting state data. An alternative would be to test the hypothesis in the movie-data residuals — for which, the most promising approach is probably to use the average time course in N-1 subjects as a measure of movie-driven activity, and then measure interregional correlations in the residuals after removing variance explained by the movie-driven timecourse.

Then I have more minor issues:

- I disagree with the characterization of children as being "without" ToM, when they failed false belief tasks, and "with" ToM otherwise. I know this is a matter of scientific taste, but to my taste (and, evidently in these data) ToM development begins before and extends after children pass false belief tasks. I would prefer the authors use different language and labels.
- I'm not a fan of DCM. I find the results in this manuscript basically uninterpretable, but I usually feel that way about DCM. But, if the authors are going to include it, at least have the order of regions always the same, in every matrix, so readers can tell by looking whether the patterns in different groups are similar or different. I honestly cannot see what aspect of these data supports the authors' claim about "inverted" connectivity.
- I would prefer the authors state in the abstract that these results are based entirely on openly accessible data — so that readers do not expect that or wonder whether the contribution of this paper includes the creation of a new dataset.

(Remarks on code availability)

Reviewer #2

(Remarks to the Author)

This work examines the role of the cerebellum in Theory of Mind processing as a function of developmental stage in children and compares the results with similar analyses in adults. The primary finding appears to be that the cerebellar circuits (Crus I-II) appear to change this activation and connectivity as children develop their social cognition skills. Overall, the manuscript is well written, thorough and presents a compelling argument for the developmental role of Crus I-II in theory of mind. Strengths appear the strong theoretical basis of the work, the main sample size and the use of complimentary data sets.

I recommend this for publication with only minor suggestions for issues to be addressed.

1. Age appears to be a confound: In the children who had ToM (passers N=22) vs those that did not (non-passers N=19) it's not clear if there is a main effect of age or sex in this contrast. Figure 1g indicates that there is an association between ToM score and age indicating that age is a confound. But it also shows that there is a group of young subjects (5-7) with ToM score of 6 and a group of older subjects 9-12 with the same ToM score – were these contrasted to examine a main effect of age?

2. Was there a group effect of motion? Were children with vs without ToM balanced in terms of motion during acquisition? It seems that the main effect of the analysis shown in Fig 1 b&c is stronger activation or better statistical power but this could simply be due to motion if this was not balanced.

3. This is an association study: As Marek has pointed out association studies are often difficult to replicate. There are also many different analyses presented here and while corrections for multiple comparisons are performed within each analysis it's not clear that there was an overall correction taking into account the number of tests. For example, line 204 performed seed-to-voxel correlation analyses – but did not convert to predictive models. Were any of these measures predictive?

4. Anatomic differences? The study compares adults to children at various points and acknowledges potential structural differences but more should be discussed in terms of how structural differences might account for some of the changes observed. For example using spherically defined ROI is generally a poor approach (as opposed to data driven ROIs) and one wonders what the effect of simple structure is in these crude volumes – particularly critical in a small region like the cerebellum with its many folds. –

5. Final comment on predictive models: While a GLM with outside of the magnet ToM performance as a predictor was performed for the task data, this model was not then shown to be predictive – which would add strength to the claims of an association. It might also be informative to do connectome based predictive modeling on the continuous data for the 122 children using the same scores (0-6) as the behavioral measured.

(Remarks on code availability)

The data appears to be available. The code is on a github site and the link is provided. I have not reviewed the code.

Reviewer #3

(Remarks to the Author)

The manuscript examines the role of the cerebellum in the development of theory of mind (ToM) in different age groups. The results show that children with emerging ToM abilities show similar activation in Crus I-II of the cerebellar as adults, whereas this activation is absent in children without ToM abilities. Impressively, the study reveals a developmental shift in connectivity directionality from upstream in childhood to downstream in adulthood, which is essential for the uninterrupted development of social cognition.

This study provides new insights into the role of the cerebellum in the development of social cognition and highlights changes in cerebellar-cerebral connectivity from childhood to adulthood. The results correspond well with existing theories and show that children with emerging ToM abilities show similar activation patterns in the cerebellum as adults, while this is not the case for children without such abilities. Furthermore, the discovery of a developmental shift in connectivity directionality from childhood to adulthood provides important insights into the mechanisms that support the ongoing development of social cognition. Overall, the study offers valuable contributions to our understanding of the neural basis of ToM development.

I focused on two critical criteria when evaluating the manuscript: 1) the potential of the method to improve our understanding of the role of the cerebellum in the development of social cognition, and 2) the reasonableness and interpretability of the findings compared to existing literature on ToM and cerebellar function. Here are my reflections on the paper:

Major points

1) My concern is primarily with the sample, as the selection of the sample in the current manuscript may affect the reliability and generalizability of the conclusions. First, the power of the sample of 41 children is limited. Furthermore, the exclusion of a large number of children (71 of 122) raises concerns about possible selection bias. It would be beneficial to discuss this limitation and its possible impact on the results. Secondly, the age range of children (3-12 years) is quite wide. Developmental changes in this time range are significant, and more detailed age stratification or analysis could provide clearer insights. The authors might consider using the Healthy Brain Network (HBN) developmental database to address these issues. This database covers a suitable age range (5-21 years) and has a large sample size. It also includes the movie "The Present," which, like "Partly Cloudy," is used to study ToM in children and adults. The HBN also collects extensive phenotypic data for each participant, including developmental status and environment, and in particular the Social Responsiveness Scale, which effectively measures understanding of mental state. Using this database and reanalyzing the data could help ensure the reliability and generalizability of the results.

2) While the introduction sets out the aim of the study, it would be helpful to explicitly state the hypothesis and specific objectives early on. Provide a more detailed background on the development of ToM in children. Discuss key milestones and the age range at which critical ToM abilities typically emerge. Highlight any gaps or inconsistencies in the literature that the current study aims to address. This will strengthen the rationale for why this study is necessary and how it builds on or departs from existing research.

Here are some more specific methodological comments in no particular order:

- The current approach in the manuscript involves separately identifying brain regions that correlate with ToM scores within each group (passers and non-passers) and then further examining differences between these groups. However, it raises the question of why the authors did not directly look for brain regions that correlate with ToM scores in the entire sample or directly identify brain regions with differences between the two groups. Specifically, what are the advantages of separately identifying correlated brain regions within each group before examining differences, compared to directly analyzing correlations and differences in brain regions across the entire sample?

- The sentence "This suggests that, even though children with ToM abilities display functional connectivity between the rCrus II and the cerebral ToM network, these connections still increase in strength until adulthood and thus have not reached full maturity by middle childhood" implies a developmental course of connectivity strength. However, this claim would benefit from a more explicit analysis. Could the authors provide a correlation analysis between connectivity strength and age? This would quantitatively support the claim that connectivity continues to increase in strength until adulthood.

Minor points

1) In the sentence "In the group of non-passers, we identified three significant clusters ... (Figure 1c) ... (Figure 1d) ..." There is an error in referencing the figures. The references to Figures 1c and 1d are reversed, leading to potential confusion for readers.

2) Figure 2c: The color scale for the "without ToM" group has a maximum value of 2.8, whereas the "with ToM" group has a maximum value of 4.5. Please use a consistent color scale for both comparisons to avoid misleading interpretations.

3) The terms "bodily modification," "bodily transformation," and "bodily pain" should be used consistently throughout the manuscript to ensure clarity and uniformity.

(Remarks on code availability)

Version 1:

Reviewer comments:

Reviewer #1

(Remarks to the Author)

The authors have satisfactorily addressed all of my comments.

(Remarks on code availability)

Reviewer #2

(Remarks to the Author)

The authors have adequately addressed my earlier concerns. I find this acceptable for publication.

(Remarks on code availability)

Reviewer #3

(Remarks to the Author)

The authors have addressed all of our concerns. Congratulations

However, we still remain concerned about the small sample size and its potential impact on the results. Additionally, comparing resting-state and movie-induced brain states might be problematic, as these conditions reflect different cognitive processes and cannot be directly equated. We recommend that the authors provide discussion on these two limitations.

(Remarks on code availability)

We have tested the python code and it seems to work well. We have not tested the matlab codes because we don't use this programme.

RESPONSE TO REVIEWERS (NCOMMS-24-23769-T)

We would like to extend our gratitude to the Editor and Reviewers for their positive evaluations, constructive comments, and for the opportunity to submit a revised manuscript. We feel that the comments and suggestions have greatly improved our manuscript. We have implemented three major changes in the manuscript following the suggestions provided by the Editor and the Reviewers. First, we have corrected contrast and connectivity analyses for age and sex to ensure that our results are not confounded by additional individual differences. We found that our results were consistent even after correcting for these variables. Second, we have addressed data quality concerns by comparing the effect of motion between Theory of Mind (ToM) passers and non-passers. We found that there was no significant difference in the number of motion artifacts between the two groups. Last, we have replicated our functional connectivity analyses in an independent resting-state dataset. We found that connectivity profiles between the cerebellar ToM clusters and the cerebral cortex were consistent in this independent dataset.

We have addressed other changes suggested by the Reviewers by providing additional supplementary figures and clarifications in the manuscript's Introduction, Methods, Results, and Discussion. Please find our detailed responses below according to the Reviewers' comments one by one. For the revised manuscript, changes are highlighted in yellow and are also listed in the response.

Reviewer #1 (Remarks to the Author):

Manoli et al analyse publicly available fMRI data from children and adults watching a short animated film, to examine a previously untested question: the link between theory of mind and cerebellar activity in children. They find that cerebellar regions in CRUS-I/II are active during mentalistic scenes in the movie, in children as in adults, and that both the location of the activity, and correlations with cortical regions, are more adult like in children who (according to independent behavioral measures) have more sophisticated theory of mind abilities. I believe this paper is a useful contribution to the literature. There has been increasing attention to and evidence for the cognitive functions of the cerebellum, and these results fit with that growing consensus.

We thank the Reviewer for their positive evaluation and constructive feedback on our manuscript!

I have a few major concerns that must be addressed before considering publication.

First, although many of the analyses compare neural activity ToM passers and failers, I cannot tell whether all of those analyses control for chronological age and fMRI data quality both which are correlated with ToM task performance and could confound measures of neural differences.

Thank you for highlighting this concern! In our original analyses, we chose not to control for the effect of age due to the collinearity between ToM emergence and age. In our view, age is not just a confounding variable but a relevant predictor given that we are interested in development. That is, we would indeed expect the effects to be age-related as both the brain and behavior fundamentally change with age and we are interested in how this age-related change is related to one another. Nevertheless, we have now repeated our contrast and functional connectivity analyses with chronological age (as well as sex) as covariates to detect the relationship between ToM abilities and cerebellar function over and above the influence of age, as requested. Our results in these supplementary analyses were largely consistent with the original analyses (see figures and text below). We have included details about the methods and results of these additional analyses in Supplementary Results and Supplementary Figures 2 and 4.

Supplementary results - Age and sex correction

*We performed additional analyses with age and sex as covariates, to ensure that the observed results are not driven by further individual differences. Age in particular was relatively highly correlated with ToM score (Spearman's $r(39) = .72, p < .001$; see **Figure 1g**), which is expected as both the brain and behavior fundamentally change in development. Nevertheless, we sought to investigate associations that remained significant over and above the effect of age in models where we directly contrasted children's ToM abilities. We first performed a GLM to assess ToM activation during the movie-watching task as a function of children's ToM abilities, while controlling for age and sex. The effect of ToM score on functional activation in the cerebellum (**Figure 1b**) were largely similar in bilateral medial Crus I (rCrus I: 19 -85 -30; lCrus I: -40 -75 -35) and Crus II (rCrus II: 42 -74 -45; lCrus II: -12 -69 -43) ($p_{\text{uncorr.}} < .001$, FDR-corrected: $q = .05$; **Supplementary figure 2**).*

*We then compared cerebro-cerebellar connectivity between ToM passers and non-passers by repeating our seed-to-voxel functional connectivity two-sample t -tests for rCrus I and rCrus II while controlling for age and sex. Consistent with our main analyses, ToM non-passers demonstrated greater connectivity with regions not overlapping with the cerebral ToM network for the rCrus I seed (identified in all children) ($p_{\text{uncorr.}} < .001$, FDR-corrected: $q = .05$; **Supplementary figure 4a**). Conversely, ToM passers demonstrated greater connectivity with clusters of the ToM network, namely the TPJ and STS, for the rCrus II seed (identified in ToM passers), in line with our main findings ($p_{\text{uncorr.}} < .001$, FDR-corrected: $q = .05$; **Supplementary figure 4b**). However, these connectivity differences were smaller than the ones observed in our main findings. Additionally, contrary to our main findings, connectivity between the rCrus II and the PreC, a core node of the cerebral ToM network, seemed to unexpectedly decrease in ToM passers (**Supplementary figure 4b**). This discrepancy could be at least partially attributed to the high collinearity between age and ToM score, which could obscure the true relationship between connectivity and ToM abilities, making it difficult to disentangle the effects of these two variables. Future studies should further investigate this by examining the mechanisms of cerebro-cerebellar connectivity in ToM development.*

Supplementary figure 2. Functional activation controlling for age and sex. Consistent with our main analyses, functional activation in Crus I/II increases as a function of ToM abilities.

Supplementary figure 4. Seed-to-voxel functional connectivity between cerebellar ToM clusters and the cerebral cortex controlling for age and sex. **A.** Connectivity for rCrus I seed, identified in all children. **B.** Connectivity for rCrus II seed, identified in ToM passers.

In the original submission, we addressed data quality by using meticulously preprocessed data which was made openly available by the authors of the original dataset (Richardson et al., 2018) and visually inspecting all scans before the analyses. Additionally, we used nuisance regressors

(number of artifact timepoints and principal component analysis (PCA)-based noise regressors; see Methods: Preprocessing, p. 20) generated in the original study as covariates in all first-level models. To further address data quality concerns, we have now performed a Welch's two-sample *t*-test to compare the number of artifact timepoints between the ToM passers and non-passers. There was no significant difference in the number of artifact timepoints, suggesting that the two groups were matched in terms of data quality: passers ($M=10.18$; $SD = 10.38$) and non-passers ($M=43.33$; $SD = 12.59$), $t(39) = 1.21$, $p = .27$ (Methods: Preprocessing, p. 21).

Second, the functional correlation analyses were conducted during the same movie as the contrast based analyses. I am concerned that this is a major confound: it is impossible to differentiate correlated spontaneous fluctuations (nominally evidence of functional connectivity between regions) and correlations that arise from the movie stimulus itself activating regions at the same time. So, the finding of different movie-evoked activity, and different functional correlations may be just two ways of describing the same data. This hypothesis would be much better tested in resting state data. An alternative would be to test the hypothesis in the movie-data residuals for which, the most promising approach is probably to use the average time course in N-1 subjects as a measure of movie-driven activity, and then measure interregional correlations in the residuals after removing variance explained by the movie-driven timecourse.

We thank the Reviewer for this thoughtful comment. As suggested, we have validated the functional correlation analyses (Figure 2 in the manuscript) in an independent dataset. We used resting-state data of 2-to-5 year-old children from the Baby Connectome Project (BCP; Howell et al., 2019). We opted for this dataset given the importance to cover the temporal window of ToM emergence early in life (see Flavell et al., 1990; Grosse Wiesmann et al., 2020), despite this dataset being relatively small in size after excluding children without an out-of-scanner ToM assessment and with excessive motion artifacts ($N = 26$; see Supplementary Methods). The BCP contains children's scores on the Children's Social Understanding Scale (CSUS), which is highly correlated with ToM task performance in development (Tahiroglu et al., 2014). We used children's CSUS scores to examine cerebro-cerebellar connectivity as a function of ToM abilities. The results were largely consistent with our main analyses. Specifically, we observed increased connectivity between cerebellar ToM clusters and the cerebral ToM network as a function of children's ToM abilities (see text and figure below). However, we also note some discrepancies with our original results, namely less pronounced connectivity with the precuneus and more connections with non-ToM regions for both seeds, which could be attributed to the smaller sample size in this analysis. We have included details about the methods of this additional analysis in Supplementary Methods and Results and Supplementary Figure 8.

Supplementary results - Functional connectivity replication analysis

We validated our functional connectivity analyses in resting-state data from an independent dataset to ensure that the results were indeed driven by cerebro-cerebellar connectivity and not correlations that arise from the movie stimulus itself. We used openly available data from the

*Baby Connectome Project (BCP; Howell et al., 2019), where ToM abilities were scored based on the Children's Social Understanding Scale (CSUS), a measure that is highly correlated with out-of-scanner ToM assessments (Tahiroglu et al., 2014). After excluding participants without CSUS scores and with excessive motion artifacts, our final sample consisted of 26 children (see **Supplementary methods**).*

*Results were largely consistent with the ones observed in main analyses. Specifically, as in the main analyses, the rCrus I seed, identified in all children from the Richardson et al. (2018) sample, was correlated with regions of the cerebral ToM network (e.g., TPJ, STS, dmPFC, vmPFC), as well as regions that do not typically belong to that network (e.g., dorsolateral PFC, cingulate, and thalamus) (group-level one-sample t-test, $p_{\text{uncorr.}} < .001$, FDR-corrected: $q = .05$; **Supplementary figure 8a, left**). GLM analyses with CSUS score as a continuous predictor demonstrated that connectivity with the non-ToM cerebral regions decreased, whereas connectivity with the PreC of the cerebral ToM network increased as a function of increasing CSUS scores ($p_{\text{uncorr.}} < .001$, FDR-corrected: $q = .05$; **Supplementary figure 8a, right**). As in the main analyses, the rCrus II seed, which was only found in ToM passers in the Richardson et al. sample, demonstrated more specific connections with the TPJ, STS, PreC, dmPFC, and vmPFC of the cerebral ToM network, and fewer connections with non-ToM regions than the rCrus I (group-level one-sample t-test, $p_{\text{uncorr.}} < .001$, FDR-corrected: $q = .05$; **Supplementary figure 8b, left**). These connections with the cerebral ToM network were more prominent as CSUS scores increased, as evidenced by a GLM analysis with CSUS score as a continuous predictor ($p_{\text{uncorr.}} < .001$, FDR-corrected: $q = .05$; **Supplementary figure 8b, right**).*

Together, results are in line with the increase in specificity with ToM-network connectivity as a function of children's ToM abilities that we found in the main study. However, these results should be interpreted with caution due to the small sample size in these analyses. In particular, even though we found convergent patterns with our main analysis, we also note some discrepancies in the observed clusters, for example less pronounced connectivity with the PreC and more connections with non-ToM regions for both seeds. Future studies should replicate these findings using larger sample sizes.

Supplementary figure 8. Seed-to-voxel functional connectivity between cerebellar ToM clusters and the cerebral cortex in an independent dataset (BCP; Howell et al., 2019). **A.** Connectivity for rCrus I seed, identified in all children in the original analyses. **B.** Connectivity for rCrus II seed, identified in ToM passers in the original analyses.

Then I have more minor issues:

- **I disagree with the characterization of children as being without ToM, when they failed false belief tasks, and with ToM otherwise. I know this is a matter of scientific taste, but to my taste (and, evidently in these data) ToM development begins before and extends after children pass false belief tasks. I would prefer the authors use different language and labels.**

We appreciate the comment and realize how our previous labelling of children’s ToM abilities was misleading. We now refer to children simply as ToM “passers” and “non-passers”, consistent with the original Richardson et al. (2018) manuscript.

- **I’m not a fan of DCM. I find the results in this manuscript basically uninterpretable, but I usually feel that way about DCM. But, if the authors are going to include it, at least have the order of regions always the same, in every matrix, so readers can tell by looking whether the patterns in different groups are similar or different. I honestly cannot see what aspect of these data supports the author’s claim about inverted connectivity.**

We thank the Reviewer for this suggestion! We have now updated Figure 3 to keep the order of regions the same in each subplot. We would also like to clarify what we mean by inverted connectivity in Figure 3. We observed significant connections *from* the cerebellum *to* the cerebral ToM regions in ToM passers, as signified by the asterisks in rows 5-8 in Figure 3a. Contrarily, in both adult samples significant connections were observed *from* the cerebral ToM regions *to* the

cerebellum, as signified by the asterisks in columns 6-7 in Figure 3d. This pattern was also observed when directly comparing adults and ToM passers in Figure 3e (see columns 6-7).

- **I would prefer the authors state in the abstract that these results are based entirely on openly accessible data so that readers do not expect that or wonder whether the contribution of this paper includes the creation of a new dataset.**

We have updated the abstract to further clarify the use of openly available functional MRI data:

Using openly available functional MRI data of children with emerging ToM abilities (N=41, age range: 3-12 years) and adults (N=78), we showed that children who passed a false-belief assessment of ToM abilities activated cerebellar Crus I-II in response to ToM events during a movie-watching task, similar to adults.

Reviewer #2 (Remarks to the Author):

This work examines the role of the cerebellum in Theory of Mind processing as a function of developmental stage in children and compares the results with similar analyses in adults. The primary finding appears to be that the cerebellar circuits (Crus I-II) appear to change this activation and connectivity as children develop their social cognition skills. Overall, the manuscript is well written, thorough and presents a compelling argument for the developmental role of Crus I-II in theory of mind. Strengths appear the strong theoretical basis of the work, the main sample size and the use of complimentary data sets.

I recommend this for publication with only minor suggestions for issues to be addressed.

We thank the Reviewer for the positive feedback on our work!

1. Age appears to be a confound: In the children who had ToM (passers N=22) vs those that did not (non=passers N=19) it's not clear if there is a main effect of age or sex in this contrast. Figure 1g indicates that there is an association between ToM score and age indicating that age is a confound. But it also shows that there is a group of young subjects (5-7) with ToM score of 6 and a group of older subjects 9-12 with the same ToM score were these contrasted to examine a main effect of age?

We thank the Reviewer for these comments. We have now corrected our contrast and functional connectivity GLM analyses for age and sex. Our results were largely consistent with the main analyses and have now included these analyses in Supplementary Figures 2 and 4. Of note, in our main analyses we do not control for age effects due to the collinearity between ToM scores and age. As ToM emergence is inherently a developmental phenomenon, we do not consider age as a confound but rather as a driving factor of the changes in ToM and in brain maturation. Nevertheless, to account for age and sex effects, we have now included age and sex as covariates in models where we directly compared ToM passers and non-passers (i.e., GLMs in contrast and

connectivity analyses) yielding results that are consistent with our main analyses (see figure and text below). We have included results of these analyses in Supplementary Results and Supplementary Figures 2 and 4.

Supplementary results - Age and sex correction

*We performed additional analyses with age and sex as covariates, to ensure that the observed results are not driven by further individual differences. Age in particular was relatively highly correlated with ToM score (Spearman's $r(39) = .72$, $p < .001$; see **Figure 1g**), which is expected as both the brain and behavior fundamentally change in development. Nevertheless, we sought to investigate associations that remained significant over and above the effect of age in models where we directly contrasted children's ToM abilities. We first performed a GLM to assess ToM activation during the movie-watching task as a function of children's ToM abilities, while controlling for age and sex. The effect of ToM score on functional activation in the cerebellum (**Figure 1b**) were largely similar in bilateral medial Crus I (rCrus I: 19 -85 -30; lCrus I: -40 -75 -35) and Crus II (rCrus II: 42 -74 -45; lCrus II: -12 -69 -43) ($p_{\text{uncorr.}} < .001$, FDR-corrected: $q = .05$; **Supplementary figure 2**).*

*We then compared cerebro-cerebellar connectivity between ToM passers and non-passers by repeating our seed-to-voxel functional connectivity two-sample t-tests for rCrus I and rCrus II while controlling for age and sex. Consistent with our main analyses, ToM non-passers demonstrated greater connectivity with regions not overlapping with the cerebral ToM network for the rCrus I seed (identified in all children) ($p_{\text{uncorr.}} < .001$, FDR-corrected: $q = .05$; **Supplementary figure 4a**). Conversely, ToM passers demonstrated greater connectivity with clusters of the ToM network, namely the TPJ and STS, for the rCrus II seed (identified in ToM passers), in line with our main findings ($p_{\text{uncorr.}} < .001$, FDR-corrected: $q = .05$; **Supplementary figure 4b**). However, these connectivity differences were smaller than the ones observed in our main findings. Additionally, contrary to our main findings, connectivity between the rCrus II and the PreC, a core node of the cerebral ToM network, seemed to unexpectedly decrease in ToM passers (**Supplementary figure 4b**). This discrepancy could be at least partially attributed to the high collinearity between age and ToM score, which could obscure the true relationship between connectivity and ToM abilities, making it difficult to disentangle the effects of these two variables. Future studies should further investigate this by examining the mechanisms of cerebro-cerebellar connectivity in ToM development.*

Supplementary figure 2. Functional activation controlling for age and sex. Consistent with our main analyses, functional activation in Crus I/II increases as a function of ToM abilities.

Supplementary figure 4. Seed-to-voxel functional connectivity between cerebellar ToM clusters and the cerebral cortex controlling for age and sex. **A.** Connectivity for rCrus I seed, identified in all children. **B.** Connectivity for rCrus II seed, identified in ToM passers.

2. Was there a group effect of motion? Were children with vs without ToM balanced in terms of motion during acquisition? It seems that the main effect of the analysis shown in Fig 1 b&c is stronger activation or better statistical power but this could simply be due to motion if this was not balanced.

Thank you for bringing this to our attention! There was no significant difference between the number of artifact timepoints in ToM passers and non-passers in a Welch's two-sample *t*-test: passers ($M = 10.18$; $SD = 10.38$) and non-passers ($M = 43.33$; $SD = 12.59$), $t(39) = 1.21$, $p = .27$ (Methods: Preprocessing, p. 20). This suggests that the two groups were balanced in terms of motion.

3. This is an association study: As Marek has pointed out association studies are often difficult to replicate. There are also many different analyses presented here and while corrections for multiple comparisons are performed within each analysis it's not clear that there was an overall correction taking into account the number of tests. For example, line 204 performed seed-to-voxel correlation analyses but did not convert to predictive models. Were any of these measures predictive?

We truly appreciate the Reviewer's insightful comments. As requested by Reviewer #1, we have replicated our functional connectivity results in an independent dataset (BCP; Howell et al., 2019), where we found largely consistent patterns with our main analyses (see figure and text below). Specifically, we observed increased connectivity between cerebellar ToM clusters and the cerebral ToM network as a function of children's ToM abilities (see text and figure below). We have included details about the methods and results of this additional analysis in Supplementary Methods and Results and Supplementary Figure 8.

Supplementary Results - Functional connectivity replication analysis

*We validated our functional connectivity analyses in resting-state data from an independent dataset to ensure that the results were indeed driven by cerebro-cerebellar connectivity and not correlations that arise from the movie stimulus itself. We used openly available data from the Baby Connectome Project (BCP; Howell et al., 2019), where ToM abilities were scored based on the Children's Social Understanding Scale (CSUS), a measure that is highly correlated with out-of-scanner ToM assessments (Tahiroglu et al., 2014). After excluding participants without CSUS scores and with excessive motion artifacts, our final sample consisted of 26 children (see **Supplementary methods**).*

*Results were largely consistent with the ones observed in main analyses. Specifically, as in the main analyses, the rCrus I seed, identified in all children from the Richardson et al. (2018) sample, was correlated with regions of the cerebral ToM network (e.g., TPJ, STS, dmPFC, vmPFC), as well as regions that do not typically belong to that network (e.g., dorsolateral PFC, cingulate, and thalamus) (group-level one-sample *t*-test, $p_{\text{uncorr.}} < .001$, FDR-corrected: $q = .05$; **Supplementary figure 8a, left**). GLM analyses with CSUS score as a continuous predictor demonstrated that connectivity with the non-ToM cerebral regions decreased, whereas connectivity with the PreC of the cerebral ToM network increased as a function of increasing CSUS scores ($p_{\text{uncorr.}} < .001$, FDR-corrected: $q = .05$; **Supplementary figure 8a, right**). As in the main analyses, the rCrus II seed, which was only found in ToM passers in the Richardson et al. sample, demonstrated more specific connections with the TPJ, STS, PreC, dmPFC, and vmPFC*

of the cerebral ToM network, and fewer connections with non-ToM regions than the rCrus I (group-level one-sample t -test, $p_{\text{uncorr.}} < .001$, FDR-corrected: $q = .05$; **Supplementary figure 8b, left**). These connections with the cerebral ToM network were more prominent as CSUS scores increased, as evidenced by a GLM analysis with CSUS score as a continuous predictor ($p_{\text{uncorr.}} < .001$, FDR-corrected: $q = .05$; **Supplementary figure 8b, right**).

Together, results are in line with the increase in specificity with ToM-network connectivity as a function of children's ToM abilities that we found in the main study. However, these results should be interpreted with caution due to the small sample size in these analyses. In particular, even though we found convergent patterns with our main analysis, we also note some discrepancies in the observed clusters, for example less pronounced connectivity with the PreC and more connections with non-ToM regions for both seeds. Future studies should replicate these findings using larger sample sizes.

Supplementary figure 8. Seed-to-voxel functional connectivity between cerebellar ToM clusters and the cerebral cortex in an independent dataset (BCP; Howell et al., 2019). **A.** Connectivity for rCrus I seed, identified in all children in the original analyses. **B.** Connectivity for rCrus II seed, identified in ToM passers in the original analyses.

As an additional point, in our study we are looking at functional activations based on an fMRI task, which we then further unpack by investigating possible connectivity mechanisms that could underlie them. Because of the use of an in-scanner task with clearly outlined conditions, we would like to argue that our study goes beyond pure association, unlike traditional brain-wide association studies (BWAS), which focus on cortical thickness and resting-state fMRI. The value of task-based fMRI studies with smaller samples is also outlined in Marek et al.'s (2022) seminal

study, where it is argued that they “frequently have increased measurement reliability and effect sizes” (p. 658).

We also appreciate the comment about the number of statistical analyses. We would like to note that most analyses, particularly the contrast analyses in separate groups of ToM passers and non-passers, happened post-hoc as a way of comprehensively unpacking and clarifying patterns observed in previous analyses, given the novel focus of our study. For example, in Figure 1a we explore the “overall” effect of ToM scenes in the cerebellum, without considering children’s ToM abilities. In Figure 1b, we then see that, when explicitly assessing ToM abilities, we find activation increases in specific parts of the posterior cerebellum. Similarly, Figures 1c-d further elucidate these ToM-specific activations by showing a clear and stark difference in activation patterns between ToM passers and non-passers. As such, not all analyses hold the same magnitude in our results. Moreover, the contrast, functional connectivity, and effective connectivity analyses answer different questions (i.e., which regions of the cerebellum are activated during a ToM task, how they interact with the cerebral cortex, and what the direction of these interactions is). For these reasons, we believe that a multiple comparisons correction across analyses is not appropriate. We would like to point out that all our analyses were pre-registered, and we control for multiple comparisons using the false discovery rate within each analysis. We have also further clarified the post-hoc nature of some of our analyses in the Results (p. 6):

To further clarify differences between children who have and have not yet developed ToM abilities, we performed additional post-hoc analyses.

Lastly, we would like to clarify that the current work indeed does not do out of sample prediction, but rather examines ToM emergence using different models, with the functional and effective connectivity approaches being post-hoc investigation of the original group differences reported. Our current sample size, in addition to the scarcity of available data with a compatible age range (3-5 years, corresponding to the ToM “emergence window”; Flavell et al., 1990; Grosse Wiesmann et al., 2020) and task conditions are currently not allowing us to make inferences about prediction. We have addressed this issue in the Discussion (p. 17), where we encourage future research on predictive mechanisms with larger sample sizes:

Our results primarily reflect associations between cerebellar function and individual differences in the development of ToM, without making claims about the prediction of ToM task performance based on cerebellar activation across different developmental or clinical contexts. Although our sample size does not currently support predictive inferences, we believe our findings provide a valuable framework for future research by identifying cerebellar regions directly associated with ToM development early in life, thereby opening exciting opportunities for predictive modeling.

4. Anatomic differences? The study compares adults to children at various points and acknowledges potential structural differences but more should be discussed in terms of how structural differences might account for some of the changes observed. For example using spherically defined ROI is generally a poor approach (as opposed to data driven ROIs) and one wonders what the effect of simple structure is in these crude volumes particularly critical in a small region like the cerebellum with its many folds.

Thank you for highlighting these possible issues. Regarding anatomic differences, we want to clarify that we followed the approach of the original study (Richardson et al., 2018) for adult-child comparisons, where all participants were normalized to the same MNI template. Similar procedures have been suggested for young children in other studies (Gweon et al., 2012; Cantlon et al., 2006; Bedny et al., 2015). Specifically for the cerebellum, recent population-level studies have used similar normalization approaches for normative growth models and adult-child comparisons in infants and young children (Kim et al., 2024; Liu et al., 2022; Lyu et al., 2024). Nevertheless, we address possible concerns about anatomical differences between adults and children in the Discussion (p. 16):

An additional consideration is possible anatomical differences in the context of adult-child comparisons. Following Richardson et al.'s²⁸ original study, we normalized child and adult brains to the same standard template. As in the original study, this was motivated by similar procedures extending to children under 7 years^{28,53}. Nevertheless, we acknowledge the existence of shape and size differences between the developing and the adult brain, which could have influenced our results.

Regarding the Reviewer's point about ROI definition, we would like to point out that our ROIs are data-driven insofar as they come from group activations within our sample (please see an example ROI in the image below). Our primary reason for defining spherical 5mm ROIs around peak coordinates for our functional connectivity analyses (as opposed to using the entire functional clusters as our ROIs) was that we wanted to remain conservative and focus on the connectivity of only peak coordinates in our functional clusters with the cerebral cortex. An additional reason for the use of spherical ROIs is large-scale evidence that functional and anatomical boundaries in the adult and the developing cerebellum are independent of each other (Buckner et al., 2011; King et al., 2019; Lyu et al., 2024; Nettekoven et al., 2024). Specifically, functional activations related to different cognitive domains (including ToM) do not seem to be bounded by the cerebellum's anatomical folds. Given the absence of clear evidence of how volume influences function in the cerebellum, we believe that the use of spherical ROIs instead of anatomically circumscribed ones are more appropriate in this case.

Example ROI (rCrus II) on the MNI152 template.

5. Final comment on predictive models: While a GLM with outside of the magnet ToM performance as a predictor was performed for the task data, this model was not then shown to be predictive; which would add strength to the claims of an association. It might also be informative to do connectome based predictive modeling on the continuous data for the 122 children using the same scores (0-6) as the behavioral measured.

We appreciate the further insight on predictive models. Unfortunately, we are not able to use the whole range of available data for 122 children. This is because only 41 children ended up having full cerebellar coverage after visual inspection of all images. As explained above, our current sample size, in addition to the lack of available data with similar parameters, do not allow us to construct predictive models. We further address this in the Discussion, where we present predictive modelling as an exciting opportunity for future research (p. 17).

Reviewer #2 (Remarks on code availability):

The data appears to be available. The code is on a github site and the link is provided. I have not reviewed the code.

Reviewer #3 (Remarks to the Author):

The manuscript examines the role of the cerebellum in the development of theory of mind (ToM) in different age groups. The results show that children with emerging ToM abilities show similar activation in Crus I-II of the cerebellum as adults, whereas this activation is absent in children without ToM abilities. Impressively, the study reveals a developmental shift in connectivity directionality from upstream in childhood to downstream in adulthood, which is essential for the uninterrupted development of social cognition.

This study provides new insights into the role of the cerebellum in the development of social cognition and highlights changes in cerebellar-cerebral connectivity from childhood to adulthood. The results correspond well with existing theories and show that children with emerging ToM abilities show similar activation patterns in the cerebellum as adults, while this is not the case for children without such abilities. Furthermore, the discovery of a developmental shift in connectivity directionality from childhood to adulthood provides important insights into the mechanisms that support the ongoing development of social cognition. Overall, the study offers valuable contributions to our understanding of the neural basis of ToM development.

I focused on two critical criteria when evaluating the manuscript: 1) the potential of the method to improve our understanding of the role of the cerebellum in the development of social cognition, and 2) the reasonableness and interpretability of the findings compared to existing literature on ToM and cerebellar function. Here are my reflections on the paper:

We appreciate the Reviewer's positive and constructive evaluation of our study!

Major points

1) My concern is primarily with the sample, as the selection of the sample in the current manuscript may affect the reliability and generalizability of the conclusions. First, the power of the sample of 41 children is limited. Furthermore, the exclusion of a large number of children (71 of 122) raises concerns about possible selection bias. It would be beneficial to discuss this limitation and its possible impact on the results. Secondly, the age range of children (3-12 years) is quite wide. Developmental changes in this time range are significant, and more detailed age stratification or analysis could provide clearer insights. The authors might consider using the Healthy Brain Network (HBN) developmental database to address these issues. This database covers a suitable age range (5-21 years) and has a large sample size. It also includes the movie "The Present," which, like "Partly Cloudy," is used to study ToM in children and adults. The HBN also collects extensive phenotypic data for each participant, including developmental status and environment, and in particular the Social Responsiveness Scale, which effectively measures understanding of mental state. Using this database and reanalyzing the data could help ensure the reliability and generalizability of the results.

We thank the Reviewer for their comments! First, we acknowledge the Reviewer's concerns about sample size limitations. The sample size was modest due to incomplete cerebellar coverage in the original dataset. The dataset was not collected with the cerebellum in mind, and parts of

this structure, particularly on the inferior side, were cropped out of the field of view for many subjects during BOLD data acquisition (please see the figure below for an example).

Example preprocessed BOLD image with incomplete coverage on the MNI152 template.

Coverage was considered complete if substantial parts of the cerebellum were not missing when the standard cerebellar atlas mask (SUIT; Diedrichsen, 2006) was overlaid on the images, therefore there was no selection bias when inspecting the images. We now address sample size concerns in the Discussion, where we acknowledge the modest size of the current sample and the need for larger samples to further investigate the cerebellar basis of ToM emergence. Nevertheless, the fact that our functional connectivity results were validated in an independent dataset increases our confidence in the generalizability of our findings (p. 16):

A further limitation is our developmental sample size. Given that most participants in the original dataset had incomplete cerebellar coverage, we limited our analyses to 41 children. Given the modest sample size in this analysis, future studies should further investigate the role of the cerebellum in ToM emergence in larger samples. Nevertheless, our functional connectivity results were validated in an independent dataset, which increases our confidence in the generalizability of results.

Regarding the Reviewer's comment about the age range, we would like to clarify that our sample was primarily composed of younger children (mean age: ~6 years), with fewer observations in the 8–12-year age range ($N = 10$). We nonetheless chose to include older children in our analyses to increase statistical power. It is important to note that age did not seem to substantially alter our

results, as evidenced by supplementary contrast and connectivity analyses (see Supplementary Figures 2 and 4, also pasted below).

Supplementary figure 2. Functional activation controlling for age and sex. Consistent with our main analyses, functional activation in Crus I/II increases as a function of ToM abilities.

Supplementary figure 4. Seed-to-voxel functional connectivity between cerebellar ToM clusters and the cerebral cortex controlling for age and sex. **A.** Connectivity for rCrus I seed, identified in all children. **B.** Connectivity for rCrus II seed, identified in ToM passers.

Lastly, we thank the Reviewer for their excellent suggestion to replicate our analyses in an open dataset. We chose to do this with the Baby Connectome Project (BCP) dataset (Howell et al., 2019) instead of the suggested HBN dataset, because the HBN dataset only covers children from 5 to 21 years, whereas our research question requires the critical period of ToM emergence at 3 to 5 years covered by the BCP (see Flavell et al., 1990 and Grosse Wiesmann et al., 2020). In addition, the BCP includes an out-of-scanner assessment of children's ToM abilities (CSUS; Tahiroglu et al., 2014). The results were largely consistent with our main findings. We found increased connectivity between ToM clusters in the cerebellum and the cerebral ToM network as a function of children's ToM abilities (see text and figure below). However, we also note some discrepancies with our original results, namely less pronounced connectivity with the precuneus and more connections with non-ToM regions for both seeds, which could be attributed to the smaller sample size in this analysis ($N = 26$), after excluding children without a CSUS assessment and with excessive motion artifacts (see Supplementary Methods). We have included details about the methods of this additional analysis in Supplementary Methods and Results and Supplementary Figure 8.

Supplementary results - Functional connectivity replication analysis

*We validated our functional connectivity analyses in resting-state data from an independent dataset to ensure that the results were indeed driven by cerebro-cerebellar connectivity and not correlations that arise from the movie stimulus itself. We used openly available data from the Baby Connectome Project (BCP; Howell et al., 2019), where ToM abilities were scored based on the Children's Social Understanding Scale (CSUS), a measure that is highly correlated with out-of-scanner ToM assessments (Tahiroglu et al., 2014). After excluding participants without CSUS scores and with excessive motion artifacts, our final sample consisted of 26 children (see **Supplementary methods**).*

*Results were largely consistent with the ones observed in main analyses. Specifically, as in the main analyses, the rCrus I seed, identified in all children from the Richardson et al. (2018) sample, was correlated with regions of the cerebral ToM network (e.g., TPJ, STS, dmPFC, vmPFC), as well as regions that do not typically belong to that network (e.g., dorsolateral PFC, cingulate, and thalamus) (group-level one-sample t-test, $p_{\text{uncorr.}} < .001$, FDR-corrected: $q = .05$; **Supplementary figure 8a, left**). GLM analyses with CSUS score as a continuous predictor demonstrated that connectivity with the non-ToM cerebral regions decreased, whereas connectivity with the PreC of the cerebral ToM network increased as a function of increasing CSUS scores ($p_{\text{uncorr.}} < .001$, FDR-corrected: $q = .05$; **Supplementary figure 8a, right**). As in the main analyses, the rCrus II seed, which was only found in ToM passers in the Richardson et al. sample, demonstrated more specific connections with the TPJ, STS, PreC, dmPFC, and vmPFC of the cerebral ToM network, and fewer connections with non-ToM regions than the rCrus I (group-level one-sample t-test, $p_{\text{uncorr.}} < .001$, FDR-corrected: $q = .05$; **Supplementary figure 8b, left**). These connections with the cerebral ToM network were more prominent as CSUS scores*

increased, as evidenced by a GLM analysis with CSUS score as a continuous predictor ($p_{\text{uncorr.}} < .001$, FDR-corrected: $q = .05$; **Supplementary figure 8b, right**).

Together, results are in line with the increase in specificity with ToM-network connectivity as a function of children's ToM abilities that we found in the main study. However, these results should be interpreted with caution due to the small sample size in these analyses. In particular, even though we found convergent patterns with our main analysis, we also note some discrepancies in the observed clusters, for example less pronounced connectivity with the PreC and more connections with non-ToM regions for both seeds. Future studies should replicate these findings using larger sample sizes.

Supplementary figure 8. Seed-to-voxel functional connectivity between cerebellar ToM clusters and the cerebral cortex in an independent dataset (BCP; Howell et al., 2019). **A.** Connectivity for rCrus I seed, identified in all children in the original analyses. **B.** Connectivity for rCrus II seed, identified in ToM passers in the original analyses.

We should note that the BCP only includes resting-state functional data, which is why it was only possible to replicate our functional connectivity analyses in this dataset. We were unfortunately not able to find other datasets with a compatible age range and task specifications. We address this in the Discussion, where we encourage future research on wider age ranges to examine the role of the cerebellum in ToM processing in different stages of development (p. 17):

Our developmental sample was primarily composed of younger children (mean age: ~6 years), with fewer observations in the 8–12-year age range ($N = 10$). The observed differences between childhood and adulthood raise the exciting question of how cerebellar activations and cerebro-cerebellar connectivity in the context of ToM change through later childhood and

adolescence until adulthood. For example, future studies could investigate cerebro-cerebellar connectivity in ToM at different stages of development to see when and how the directionality of cerebro-cerebellar switches from an upstream model in childhood to a downstream model in adulthood.

2) While the introduction sets out the aim of the study, it would be helpful to explicitly state the hypothesis and specific objectives early on. Provide a more detailed background on the development of ToM in children. Discuss key milestones and the age range at which critical ToM abilities typically emerge. Highlight any gaps or inconsistencies in the literature that the current study aims to address. This will strengthen the rationale for why this study is necessary and how it builds on or departs from existing research.

We appreciate the Reviewer's constructive feedback on the Introduction! We have now substantially expanded this section to provide additional background on the development of ToM, including key milestones for its emergence (p. 3-4):

In development, the emergence of ToM abilities has been extensively studied across populations ranging from infancy to late childhood²⁸. A pivotal milestone in the development of ToM reasoning occurs between the ages of 3 and 5, a “breakthrough” period in which children typically start succeeding in false-belief tasks, widely regarded as a critical test of ToM abilities²⁶. These tasks require children to recognize false beliefs held by a story character, typically in the context of the character's mental misrepresentations regarding an object's location, content, or nature^{26,55,77}. Successfully passing false-belief tasks is argued to reflect the emergence of representations of others' mental states⁷⁸. On a neural level, this cognitive breakthrough is supported by a rich spectrum of structural and functional changes within the cerebral ToM network. Success in false-belief tasks during this developmental period is associated with increases in cortical thickness, surface area, and white matter maturation of regions of the ToM network^{3,27,77}. These structural changes are accompanied by greater functional specialization and strengthened connectivity among regions of the ToM network as a function of children's emerging ToM abilities^{28,77}.

We have also further clarified literature gaps that our work addresses (p. 3):

Recent advancements in functional magnetic resonance imaging (fMRI) have greatly enhanced our neurobiological understanding of ToM, revealing its neural correlates not only in the cerebral cortex^{3,4,5} but also in the adult cerebellum^{6,7,8,9}. Although substantial evidence highlights the cerebellum's involvement in ToM processing in adults, its role in the development of ToM remains poorly understood. This gap is particularly striking given that cerebellar lesions acquired during development are linked to profound and enduring ToM deficits—far more severe than those observed following lesions sustained in adulthood^{10,11,12}—suggesting a crucial

involvement of the developing cerebellum in ToM understanding. To address this issue, the present study investigated the functional contribution of the cerebellum to the development of ToM during early childhood to better understand how this structure is associated with the emergence of ToM abilities.

Here are some more specific methodological comments in no particular order:

- The current approach in the manuscript involves separately identifying brain regions that correlate with ToM scores within each group (passers and non-passers) and then further examining differences between these groups. However, it raises the question of why the authors did not directly look for brain regions that correlate with ToM scores in the entire sample or directly identify brain regions with differences between the two groups. Specifically, what are the advantages of separately identifying correlated brain regions within each group before examining differences, compared to directly analyzing correlations and differences in brain regions across the entire sample?

We apologize for the confusion. Our primary analyses indeed involved ToM associations in the entire sample, either without or with children's false belief scores as a continuous predictor (Figure 1a and Figure 1b, respectively). Figure 1c and Figure 1d then unpack the observed differences in Figure 1b further by examining the groups of ToM passers and non-passers separately. We have now further clarified this in the Results section (p. 5-6).

- The sentence "This suggests that, even though children with ToM abilities display functional connectivity between the rCrus II and the cerebral ToM network, these connections still increase in strength until adulthood and thus have not reached full maturity by middle childhood" implies a developmental course of connectivity strength. However, this claim would benefit from a more explicit analysis. Could the authors provide a correlation analysis between connectivity strength and age? This would quantitatively support the claim that connectivity continues to increase in strength until adulthood.

We thank the Reviewer for this insightful comment. Most children in our sample are between 3-7 years (see also Figure 1g). Because of this, and because of the lack of continuous data that covers the entire developmental spectrum until adulthood, it would be difficult to determine whether cerebro-cerebellar connections increase with age with a correlational analysis. Thus, we realized that our claim was unfounded and have rephrased this sentence in the Results (p. 9):

This suggests that, even though children with ToM abilities display functional connectivity between the rCrus II and the cerebral ToM network, these connections have not yet reached full maturity.

Minor points

1) In the sentence "In the group of non-passers, we identified three significant clusters (Figure 1c) (Figure 1d) ..." There is an error in referencing the figures. The references to Figures 1c and 1d are reversed, leading to potential confusion for readers.

Thank you for bringing this to our attention! We have now fixed the figure references in the text.

2) Figure 2c: The color scale for the "without ToM" group has a maximum value of 2.8, whereas the "with ToM" group has a maximum value of 4.5. Please use a consistent color scale for both comparisons to avoid misleading interpretations.

We have homogenized our color scales in this figure to avoid this issue.

3) The terms "bodily modification," "bodily transformation," and "bodily pain" should be used consistently throughout the manuscript to ensure clarity and uniformity.

We now consistently refer to the associated condition as "bodily pain" to ensure clarity in the manuscript.

RESPONSE TO REVIEWERS (NCOMMS-24-23769-A)

We would like to once again thank the Editor and Reviewers for their positive evaluations of our manuscript and the opportunity to publish our work in *Nature Communications*. Here, we address some final comments made by the Reviewers.

Reviewer #1

The authors have satisfactorily addressed all of my comments.

We thank the Reviewer for their positive evaluation of our work!

Reviewer #2

The authors have adequately addressed my earlier concerns. I find this acceptable for publication.

We thank the Reviewer for their positive evaluation of our work!

Reviewer #3

The authors have addressed all of our concerns. Congratulations

However, we still remain concerned about the small sample size and its potential impact on the results. Additionally, comparing resting-state and movie-induced brain states might be problematic, as these conditions reflect different cognitive processes and cannot be directly equated. We recommend that the authors provide discussion on these two limitations.

We thank the Reviewer for their positive evaluation and additional feedback! We have addressed the Reviewer's concerns in the limitations section of the Discussion:

A further limitation is our developmental sample size. Given that most participants in the original dataset had incomplete cerebellar coverage, we limited our analyses to 41 children. Given the modest sample size in this analysis, future studies should further investigate the role of the cerebellum in ToM emergence in larger samples. Additionally, the use of both resting-state and movie-induced brain states in our validation analysis may present interpretational challenges, as these conditions engage distinct cognitive processes. While we used both states to investigate functional connectivity patterns, we acknowledge that they are not directly equivalent and may tap into different aspects of neural function. Nevertheless, the consistency of findings across the movie-based primary dataset and the resting-state validation dataset strengthens our confidence in the generalizability of the results.